# Sketch-GNN: Scalable Graph Neural Networks with Sublinear Training Complexity

**Mucong Ding , Tahseen Rabbani , Bang An , Evan Z Wang , Furong Huang**

Department of Computer Science, University of Maryland
*{mcding, trabbani, bangan, furongh}@cs.umd.edu*

## Abstract

Graph Neural Networks (GNNs) are widely applied to graph learning problems such as node classification. When scaling up the underlying graphs of GNNs to a larger size, we are forced to either train on the complete graph and keep the full graph adjacency and node embeddings in memory (which is often infeasible) or mini-batch sample the graph (which results in exponentially growing computational complexities with respect to the number of GNN layers). Various sampling-based and historical-embedding-based methods are proposed to avoid this exponential growth of complexities. However, none of these solutions eliminates the linear dependence on graph size. This paper proposes a sketch-based algorithm whose training time and memory grow sublinearly with respect to graph size by training GNNs atop a few compact sketches of graph adjacency and node embeddings. Based on polynomial tensor-sketch (PTS) theory, our framework provides a novel protocol for sketching non-linear activations and graph convolution matrices in GNNs, as opposed to existing methods that sketch linear weights or gradients in neural networks. In addition, we develop a locality sensitive hashing (LSH) technique that can be trained to improve the quality of sketches. Experiments on large-graph benchmarks demonstrate the scalability and competitive performance of our Sketch-GNNs versus their full-size GNN counterparts.

## 1 Introduction

Graph Neural Networks (GNNs) have achieved state-of-the-art graph learning in numerous applications, including classification [27], clustering [3], recommendation systems [43], social networks [16] and more, through representation learning of target nodes using information aggregated from neighborhoods in the graph. The manner in which GNNs utilize graph topology, however, makes it challenging to scale learning to larger graphs or deeper models with desirable computational and memory efficiency. Full-batch training that stores the Laplacian of the complete graph suffers from a memory complexity of $\mathcal{O}(m + ndL + d^2L)$ on an $n$-node, $m$-edge graph with node features of dimension $d$ when employing an $L$-layer graph convolutional network (GCN). This linear memory complexity dependence on $n$ and the limited memory capacity of GPUs make it difficult to train on large graphs with millions of nodes or more. As an example, the *MAG240M-LSC* dataset [21] is a node classification benchmark with over 240 million nodes that takes over 202 GB of GPU memory when fully loaded.

To address the memory constraints, two major lines of research are proposed: (1) Sampling-based approaches [18, 11, 12, 14, 45] based on the idea of implementing message passing only between the neighbors within a sampled mini-batch; (2) Historical-embedding based techniques, such as GNNAutoScale [17] and VQ-GNN [15]), which maintain the expressive power of GNNs on sampled subgraphs using historical embeddings. However, all of these methods require the number of mini-

36th Conference on Neural Information Processing Systems (NeurIPS 2022).

batches to be proportional to the size of the graph for fixed memory consumption. In other words, they significantly increase computational time complexity in exchange for memory efficiency when scaling up to large graphs. For example, training a 4-layer GCN with just 333K parameters (1.3 MB) for 500 epochs on *ogbn-papers100M* can take more than 2 days on a powerful AWS p4d.24x large instance [21].

We seek to achieve efficient training of GNNs with time and memory complexities sublinear in graph size without significant accuracy degradation. Despite the difficulty of this goal, it should be achievable given that (1) the number of learnable parameters in GNNs is independent of the graph size, and (2) training may not require a traversal of all local neighborhoods on a graph but rather only the most representative ones (thus sublinear in graph size) as some neighborhoods may be very similar. In addition, commonly-used GNNs are typically small and shallow with limited model capacity and expressive power, indicating that a modest proportion of data may suffice.

This paper presents *Sketch-GNN*, a framework for training GNNs with sublinear time and memory complexity with respect to graph size. Using the idea of sketching, which maps high-dimensional data structures to a lower dimension through entry hashing, we sketch the $n \times n$ adjacency matrix and the $n \times d$ node feature matrix to a few $c \times c$ and $c \times d$ sketches respectively before training, where $c$ is the sketch dimension. While most existing literature focuses on sketching linear weights or gradients, we introduce a method for sketching non-linear activation units using polynomial tensor sketch theory [19]. This preserves prediction accuracy while avoiding the need to "unsketch" back to the original high dimensional graph-node space $n$, thereby eliminating the dependence of training complexity on the underlying graph size $n$. Moreover, we propose to learn and update the sketches in an online manner using learnable locality-sensitive hashing (LSH) [9]. This reduces the performance loss by adaptively enhancing the sketch quality while incurring minor overhead sublinear in the graph size. In practice, we find that the sketch-ratio $c/n$ required to maintain "full-graph" model performance drops as $n$ increases; as a result, our Sketch-GNN enjoys sublinear training scalability.

*Sketch-GNN* applies sketching techniques to GNNs to achieve training complexity sublinear to the *data* size. This is fundamentally different from the few existing works which sketch the weights or gradients [30, 13, 26, 29, 37] to reduce the memory footprint of the model and speed up optimization. To the best of our knowledge, *Sketch-GNN* is the first sub-linear complexity training algorithm for GNNs, based on LSH and tensor sketching. The sublinear efficiency obtained applies to various types of GNNs, including GCN [27] and GraphSAGE [18]. Compared to the data compression approach [22, 23], which compresses the input graph to a smaller one with fewer nodes and edges before training, our Sketch-GNN is advantageous since it does not suffer from an extremely long preprocessing time (which renders the training speedups meaningless) and performs much better across GNN types/architectures.

The remainder of this paper is organized as follows. Section 2 summarizes the notions and preliminaries of GNNs and sketching. Section 3 describes how to approximate the GNN operations on the full graph topology with sketches. Section 3.3 introduces potential drawbacks of using fixed sketches and develops algorithms for updating sketches using learnable LSHs. In Section 4, we compare our approach to the graph compression approach and other GNN scalability methods. In Section 5, we report the performance and efficiency of *Sketch-GNNs* as well as several proof-of-concept and ablation experiments. Finally, Section 6 concludes this paper with a summary of limitations, future directions, and broader impacts.

## 2   Preliminaries

**Basic Notations.** Consider a graph with $n$ nodes and $m$ edges. Connectivity is given by the adjacency matrix $A \in \{0, 1\}^{n \times n}$ and features on nodes are represented by the matrix $X \in \mathbb{R}^{n \times d}$, where $d$ is the number of features. Given a matrix $C$, let $C_{i,j}$, $C_{i,:}$, and $C_{:,j}$ denote its $(i, j)$-th entry, $i$-th row, and $j$-th column, respectively. $\odot$ denotes the element-wise (Hadamard) product, whereas $C^{\odot k}$ represents the $k$-th order element-wise power. $\| \cdot \|_F$ is the symbol for the Frobenius norm. $I_n \in \mathbb{R}^{n \times n}$ denotes the identity matrix, whereas $\mathbf{1}_n \in \mathbb{R}^n$ is the vector whose elements are all ones. $\text{Med}\{\cdot\}$ represents the element-wise median over a set of matrices. Superscripts are used to indicate multiple instances of the same kind of variable; for instance, $X^{(l)} \in \mathbb{R}^{n \times d_l}$ are the node representations on layer $l$.

**Unified Framework of GNNs.** A *Graph Neural Network (GNN)* layer receives the node representation of the preceding layer $X^{(l)} \in \mathbb{R}^{n \times d}$ as input and outputs a new representation $X^{(l+1)} \in \mathbb{R}^{n \times d}$,

where $X = X^{(0)} \in \mathbb{R}^{n \times d}$ are the input features. Although GNNs are designed following different guiding principles, such as neighborhood aggregation (GraphSAGE), spatial convolution (GCN), self-attention (GAT), and Weisfeiler-Lehman (WL) alignment (GIN [44]), the great majority of GNNs can be interpreted as performing message passing on node features, followed by feature transformation and an activation function. The update rule of these GNNs can be summarized as [15]

$$X^{(l+1)} = \sigma\Big( \sum_q C^{(q)} X^{(l)} W^{(l,q)} \Big). \tag{1}$$

Where $C^{(q)} \in \mathbb{R}^{n \times n}$ denotes the $q$-th convolution matrix that defines the message passing operator, $q \in \mathbb{Z}_+$ is index of convolution, $\sigma(\cdot)$ is some choice of nonlinear activation function, and $W^{(l,q)} \in \mathbb{R}^{d_l \times d_{l+1}}$ denotes the learnable linear weight matrix for the $l$-th layer and $q$-th filter. GNNs under this paradigm differ from each other by their choice of convolution matrices $C^{(q)}$, which can be either fixed (GCN and GraphSAGE) or learnable (GAT). In Appendix A.1, we re-formulate a number of well-known GNNs under this framework. Unless otherwise specified, we assume $q = 1$ and $d = d_l$ for every layer $l \in [L]$ for notational convenience.

**Count Sketch and Tensor Sketch.** **(1)** *Count sketch* [7, 41] is an efficient dimensionality reduction method that projects an $n$-dimensional vector $\boldsymbol{u}$ into a smaller $c$-dimensional space using a random hash table $h : [n] \to [c]$ and a binary Rademacher variable $s : [n] \to \{\pm 1\}$, where $[n] = \{1, \ldots, n\}$. Count sketch is defined as $\mathsf{CS}(\boldsymbol{u})_i = \sum_{h(j)=i} s(j) \boldsymbol{u}_j$, which is a linear transformation of $\boldsymbol{u}$, i.e., $\mathsf{CS}(\boldsymbol{u}) = R\boldsymbol{u}$. Here, $R \in \mathbb{R}^{c \times n}$ denotes the so-called *count sketch matrix*, which has exactly one non-zero element per column. **(2)** *Tensor sketch* [32] is proposed as a generalization of count sketch to the tensor product of vectors. Given $\boldsymbol{z} \in \mathbb{R}^n$ and an order $k$, consider a $k$ number of i.i.d. hash tables $h^{(1)}, \ldots, h^{(k)} : [n] \to [c]$ and i.i.d. binary Rademacher variables $s^{(1)}, \ldots, s^{(k)} : [n] \to \{\pm 1\}$. Tensor sketch also projects vector $\boldsymbol{z} \in \mathbb{R}^n$ into $\mathbb{R}^c$, and is defined as $\mathsf{TS}_k(\boldsymbol{z})_i = \sum_{h(j_1, \cdots, j_k)=i} s^{(1)}(j_1) \cdots s^{(k)}(j_k) \boldsymbol{z}_{j_1} \cdots \boldsymbol{z}_{j_k}$, where $h(j_1, \cdots, j_k) = (h^{(1)}(j_1) + \cdots + h^{(k)}(j_k)) \bmod c$. By definition, a tensor sketch of order $k = 1$ degenerates to count sketch; $\mathsf{TS}_1(\cdot) = \mathsf{CS}(\cdot)$. **(3)** We define *count sketch of a matrix* $U \in \mathbb{R}^{d \times n}$ as the count sketch of each row vector individually, i.e., $\mathsf{CS}(U) \in \mathbb{R}^{d \times c}$ where $[\mathsf{CS}(U)]_{i,:} = \mathsf{CS}(U_{i,:})$. The *tensor sketch of a matrix* is defined in the same way. Pham and Pagh [32] devise a fast computation of tensor sketch of $U \in \mathbb{R}^{d \times n}$ (sketch dimension $c$ and order $k$) using count sketches and the Fast Fourier Transform (FFT):

$$\mathsf{TS}_k(U) = \mathsf{FFT}^{-1}\Big( \bigodot_{p=1}^k \mathsf{FFT}\big(\mathsf{CS}^{(p)}(U)\big) \Big), \tag{2}$$

where $\mathsf{CS}^{(p)}(\cdot)$ is the count sketch with hash function $h^{(p)}$ and Rademacher variable $s^{(p)}$. $\mathsf{FFT}(\cdot)$ and $\mathsf{FFT}^{-1}(\cdot)$ are the FFT and its inverse applied to each row of a matrix.

**Locality Sensitive Hashing.** The definition of count sketch and tensor sketch is based on hash table(s) that only requires a data independent uniformity, i.e., with high probability the hash-buckets are of similar size. In contrast, locality sensitive hashing (LSH) is a hashing scheme that uses locality-sensitive hash function $H : \mathbb{R}^d \to [c]$ to ensure that nearby vectors are hashed into the same bucket (out of $c$ buckets in total) with high probability while distant ones are not. *SimHash* achieves the locality-sensitive property by employing random projections [8]. Given a random matrix $P \in \mathbb{R}^{c/2 \times d}$, SimHash defines a locality-sensitive hash function

$$H(\boldsymbol{u}) = \arg\max\big([P\boldsymbol{u} \,\|\, -P\boldsymbol{u}]\big), \tag{3}$$

where $[\cdot \,\|\, \cdot]$ denotes concatenation of two vectors and $\arg\max$ returns the index of the largest element. SimHash is efficient for large batches of vectors [1]. In this paper, we apply a learnable version of SimHash that is proposed by Chen et al. [9], in which the projection matrix $P$ is updated using gradient descent; see Section 3.3 for details.

## 3 Sketch-GNN Framework via Polynomial Tensor Sketch

**Problem and Insights.** We intend to develop a "sketched counterpart" of GNNs, where training is based solely on (dimensionality-reduced) compact sketches of the convolution and node feature matrices, the sizes of which can be set independently of the graph size $n$. In each layer, Sketch-GNN

receives some sketches of the convolution matrix $C$ and node representation matrix $X^{(l)}$ and outputs some sketches of the node representations $X^{(l+1)}$. As a result, the memory and time complexities are inherently independent of $n$. The bottleneck of this problem is estimating the nonlinear activated product $\sigma(CX^{(l)}W^{(l)})$, where $W^{(l)}$ is the learnable weight of the $l$-th layer.

Before considering the nonlinear activation, as a first step, we approximate the linear product $CX^{(l)}W^{(l)}$, using dimensionality reduction techniques such as random projections and low-rank decompositions. As a direct corollary of the (distributional) Johnson–Lindenstrauss (JL) lemma [25], there exists a projection matrix $R \in \mathbb{R}^{c \times n}$ such that $CX^{(l)}W^{(l)} \approx (CR^{\mathsf{T}})(RX^{(l)}W^{(l)})$ [15]. Tensor sketch is one of the techniques that can achieve the JL bound [2]; for an error bound, see Lemma 1 in Appendix B.

Count sketch offers a good estimation of a matrix product, $CX^{(l)}W^{(l)} \approx \mathsf{CS}(C)\mathsf{CS}((X^{(l)}W^{(l)})^{\mathsf{T}})^{\mathsf{T}}$. While tensor sketch can be used to approximate the power of matrix product, i.e., $(CX^{(l)}W^{(l)})^{\odot k} \approx \mathsf{TS}_k(C)\mathsf{TS}_k((X^{(l)}W^{(l)})^{\mathsf{T}})^{\mathsf{T}}$, where $(\cdot)^{\odot k}$ is the $k$-th order element-wise power. If we combine the estimators of element-wise powers of $CX^{(l)}W^{(l)}$, we can approximate the (element-wise) activation $\sigma(\cdot)$ on $CX^{(l)}W^{(l)}$. This technique is known as a *polynomial tensor sketch (PTS)* and is discussed in [19]. In this paper, we apply PTS to sketch the message passing of GNNs, including the nonlinear activations.

## 3.1 Sketch-GNN: Approximated Update Rules

**Polynomial Tensor Sketch.** Our goal is to approximate the update rule of GNNs (Eq. (1)) in each layer. We first expand the element-wise non-linearity $\sigma$ as a power series, and then approximate the powers using count/tensor sketch, i.e.,

$$X^{(l+1)} = \sigma(CX^{(l)}W^{(l)}) \approx \sum_{k=1}^{r} c_k \left(CX^{(l)}W^{(l)}\right)^{\odot k} \approx \sum_{k=1}^{r} c_k \, \mathsf{TS}_k(C) \, \mathsf{TS}_k\left((X^{(l)}W^{(l)})^{\mathsf{T}}\right)^{\mathsf{T}},$$
(4)

where the $k = 0$ term always evaluates to zero as $\sigma(0) = 0$. In Eq. (4), coefficients $c_k$ are introduced to enable learning or data-driven selection of the weights when combing the terms of different order $k$. This allows for the approximation of a variety of nonlinear activation functions, such as sigmoid and ReLU. The error of this approximation relies on the precise estimation of the coefficients $\{c_k\}_{k=1}^{r}$. To identify the coefficients, Han et al. [19] design a coreset-based regression algorithm, which requires at least $O(n)$ additional time and memory. Since the coefficients $\{c_k\}_{k=1}^{r}$ that achieve the best performance for the classification tasks do not necessarily approximate a known activation, we propose learning the coefficients $\{c_k\}_{k=1}^{r}$ to optimize the classification loss directly using gradient descent with simple $L_2$ regularization. Experiments indicate that the learned coefficients can approximate the sigmoid activation with relative errors comparable to those of the coreset-based method; see Fig. 1a in Section 5.

**Approximated Update Rules.** The remaining step is to approximate the operations of GNNs using PTS (Eq. (4)) on sketches of convolution matrix $C$ and node representation matrix $X^{(l)}$. Consider $r$ pairwise-independent count sketches $\{\mathsf{CS}^{(k)}(\cdot)\}_{k=1}^{r}$ with sketch dimension $c$, associated with hash tables $h^{(1)}, \ldots, h^{(r)}$ and binary Rademacher variables $s^{(1)}, \ldots, s^{(r)}$, defined prior to training an $L$-layer *Sketch-GNN*. Using these hash tables and Rademacher variables, we may also construct tensor sketches $\{\mathsf{TS}_k(\cdot)\}_{k=2}^{r}$ up to the maximum order $r$.

In *Sketch-GNN*, sketches of node representations (instead of the $O(n)$ standard representation) are propagated between layers. To get rid of the dependence on $n$, we count sketch both sides of Eq. (4)

$$\begin{aligned}
S_X^{(l+1,k')} := \mathsf{CS}^{(k')}\left((X^{(l+1)})^{\mathsf{T}}\right) &\approx \mathsf{CS}^{(k')}\left(\sum_{k=1}^{r} c_k^{(l)} \mathsf{TS}_k\left((X^{(l)}W^{(l)})^{\mathsf{T}}\right)\mathsf{TS}_k(C)^{\mathsf{T}}\right) \\
&= \sum_{k=1}^{r} c_k^{(l)} \, \mathsf{TS}_k\left((X^{(l)}W^{(l)})^{\mathsf{T}}\right)\mathsf{CS}^{(k')}\left(\mathsf{TS}_k(C)^{\mathsf{T}}\right) \\
&= \sum_{k=1}^{r} c_k^{(l)} \mathsf{FFT}^{-1}\left(\bigodot_{p=1}^{k} \mathsf{FFT}\left((W^{(l)})^{\mathsf{T}} S_X^{(l,p)}\right)\right) S_C^{(l,k,k')},
\end{aligned}$$
(5)

where $S_X^{(l+1,k')} = \mathsf{CS}^{(k')}((X^{(l+1)})^{\mathsf{T}}) \in \mathbb{R}^{d \times c}$ is the transpose of column-wise count sketch of $X^{(l+1)}$, and the superscripts of $S_X^{(l+1,k')}$ indicate that it is the $k'$-th count sketch of $X^{(l+1)}$ (i.e.,

sketched by $CS^{(k)}(\cdot)$). In the second line of Eq. (5), we can move the matrix, $c_k^{(l)} \mathsf{TS}_k((X^{(l)}W^{(l)})^{\mathsf{T}})$, multiplied on the left to $\mathsf{TS}_k(C)^{\mathsf{T}}$ out of the count sketch function $CS^{(k')}(\cdot)$, since the operation of row-wise count sketch $CS^{(k')}(\cdot)$ is equivalent to multiplying the associated count sketch matrix $R^{(k')}$ on the right, i.e., for any $U \in \mathbb{R}^{n \times n}$, $CS^{(k')}(U) = UR^{(k')}$. In the third line of Eq. (5), we denote the "two-sided sketch" of the convolution matrix as $S_C^{(l,k,k')} := CS^{(k')}(\mathsf{TS}_k(C)^{\mathsf{T}}) \in \mathbb{R}^{c \times c}$ and expand the tensor sketch $\mathsf{TS}_k((X^{(l)}W^{(l)})^{\mathsf{T}})$ using the FFT-based formula (Eq. (2)).

Eq. (5) is the (recursive) **update rule** of *Sketch-GNN*, which approximates the operation of the original GNN and learns the sketches of representations. Looking at the both ends of Eq. (5), we obtain a formula that approximates the sketches of $X^{(l+1)}$ using the sketches of $X^{(l)}$ and $C$, with learnable weights $W^{(l)} \in \mathbb{R}^{d \times d}$ and coefficients $\{c_k^{(l)} \in \mathbb{R}\}_{k=1}^r$. In practice, to mitigate the error accumulation when propagating through multiple layers, we employ skip-connections across layers in Sketch-GNNs (Eq. (5) and their full-size GNN counterparts. The forward-pass and backward-propagation between the input sketches $\{S_X^{(0,k)}\}_{k=1}^r$ and the sketches of the final layer representations $\{S^{(L,k)}\}_{k=1}^r$ take $O(c)$ time and memory (see Section 3.3 for complexity details).

### 3.2 Error Bound on Estimated Representation

Based on Lemma 1 and the results in [19], we establish an error bound on the estimated final layer representation $\widetilde{X}^{(L)}$ for GCN; see Appendix B for the proof and discussions.

**Theorem 1.** *For a Sketch-GNN with $L$ layers, the estimated final layer representation is $\widetilde{X}^{(L)} = \mathsf{Med}\{R^{(k)}S_X^{(L,k)} \mid k = 1, \cdots, r\}$, where the sketches are recursively computed using Eq. (5). For $\Gamma^{(l)} = \max\{5\|X^{(l)}W^{(l)}\|_F^2, (2+3^r)\sum_i(\sum_j[X^{(l)}W^{(l)}]_{i,j})^r\}$, it holds that $\mathbf{E}(\|X^{(L)} - \widetilde{X}^{(L)}\|_F^2)/\|X^{(L)}\|_F^2 \leq \prod_{l=1}^L(1 + 2/(1 + c\lambda^{(l)^2}/nr\Gamma^{(l)})) - 1$, where $\lambda^{(l)} \geq 0$ is the smallest singular value of the matrix $Z \in \mathbb{R}^{nd \times r}$ and $Z_{:,k}$ is the vectorization of $(CX^{(l)}W^{(l)})^{\odot k}$. Moreover, if $(c(\lambda^{(l)})^2/nr\Gamma^{(l)}) \gg 1$ holds true for every layer, the relative error is $O(L(n/c))$, which is proportional to the depth of the model, and inversely proportional to the sketch ratio $(c/n)$.*

**Remarks.** Despite the fact that in Theorem 1 the error bound grows for smaller sketch ratios $c/n$, we observe in experiments that the sketch-ratio required for competitive performance decreases as $n$ increases; see Section 5. As for the number of independent sketches $r$, we know from Lemma 1 that the dependence of $r$ on $n$ is $r = \Omega(3^{\log_c n})$ which is negligible when $n$ is not too small; thus, in practice $r = 3$ is used.

The theoretical framework may not completely correspond to reality. Experimentally, the coefficients $\{\{c_k^{(l)}\}_{k=1}^r\}_{l=1}^L$ with the highest performance do not necessarily approximate a known activation. We defer the challenging problem of bounding the error of sketches and coefficients learned by gradients to future studies. Although the error bound is in expectation, we do not train over different sketches per iteration due to the instability caused by randomness. Instead, we introduce learnable locality sensitive hashing (LSH) in the next section to counteract the approximation limitations caused by the fixed number of sketches.

### 3.3 A Practical Implementation: Learning Sketches using LSH

**Motivations of Learnable Sketches.** In Section 3, we apply polynomial tensor sketch (PTS) to approximate the operations of GNNs on sketches of the convolution and feature matrices. Nonetheless, the pre-computed sketches are fixed during training, resulting in **two major drawbacks**: **(1)** The performance is limited by the quality of the initial sketches. For example, if the randomly-generated hash tables $\{h^{(k)}\}_{k=1}^r$ have unevenly distributed buckets, there will be more hash collisions and consequently worse sketch representations. The performance will suffer because only sketches are used in training. **(2)** More importantly, when multiple Sketch-GNN layers are stacked, the input representation $X^{(l)}$ changes during training (starting from the second layer). Fixed hash tables are not tailored to the "changing" hidden representations.

We seek a method for efficiently constructing high-quality hash tables tailored for each hidden embedding. Locality sensitive hashing (LSH) is a suitable tool since it is data-dependent and

preserves data similarity by hashing similar vectors into the same bucket. This can significantly improve the quality of sketches by reducing the errors due to hash collisions.

**Combining LSH with Sketching.** At the time of sketching, the hash table $h^{(k)} : [n] \to [c]$ is replaced with an LSH function $H^{(k)} : \mathbb{R}^d \to [c]$, for any $k \in [r]$. Specifically, in the $l$-th layer of a Sketch-GNN, we hash the $i$-th node to the $H^{(k)}(X_{i,:}^{(l)})$-th bucket for every $i \in [n]$, where $X_{i,:}^{(l)}$ is the embedding vector of node $i$. As a result, we define a data-dependent hash table

$$h^{(l,k)}(i) = H^{(k)}(X_{i,:}^{(l)}) \tag{6}$$

that can be used for computing the sketches of $S_X^{(l,k)}$ and $S_C^{(l,k,k')}$. This LSH-based sketching can be directly applied to sketch the fixed convolution matrix and the input feature matrix. If SimHash is used, i.e., $H^{(k)}(\boldsymbol{u}) = \arg\max \left( \left[ P^{(k)}\boldsymbol{u} \parallel -P^{(k)}\boldsymbol{u} \right] \right)$ (Eq. (3)), an additional $O(ncr(\log c + d))$ computational overhead is introduced to hash the $n$ nodes for the $r$ hash tables during preprocessing; see Appendix F more information. SimHash(es) are implemented as simple matrix multiplications that are practically very fast.

In order to employ LSH-based hash functions customized to each layer to sketch the hidden representations of a Sketch-GNN (i.e., $l = 2, \ldots, L-1$), we face **two major challenges**: **(1)** Unless we explicitly unsketch in each layer, the estimated hidden representations $\widetilde{X}^{(l)}(l = 2, \ldots, L-1)$ cannot be accessed and used to compute the hash tables. However, unsketching any hidden representation, i.e., $\widetilde{X}^{(l)} = \mathsf{Med}\{R^{(k)}S_X^{(l,k)} \mid k = 1, \cdots, r\}$, requires $O(n)$ memory and time. We need to come up with an efficient algorithm that updates the hash tables without having to unsketch the complete representation. **(2)** It's unclear how to change the underlying hash table of a sketch across layers without unsketching to the $n$-dimensional space, even if we know the most up-to-date hash tables suited to each layer.

The **challenge (2)**, i.e., changing the underlying hash table of across layers, can be solved by maintaining a sparse $c \times c$ matrix $T^{(l,k)} := R^{(l,k)}(R^{(l+1,k)})^{\mathsf{T}}$ for each $k \in [r]$, which only requires $O(cr)$ memory and time overhead; see Appendix C for more information and detailed discussions. We focus on **challenge (1)** for the remainder of this section.

**Online Learning of Sketches.** To learn a hash table tailored for a hidden layer using LSH without unsketching, we develop an efficient algorithm to update the LSH function using only a size-$|B|$ subset of the length-$n$ unsketched representations, where $B$ denotes a subset of nodes we select. This algorithm, which we term *online learning of sketches*, is made up of two key parts: *(Part 1)* select a subset of nodes $B \subseteq [n]$ to effectively update the hash table, and *(Part 2)* update the LSH function $H(\cdot)$ with a triplet loss computed using this subset.

*(1) Selection of subset $B$:* Because model parameters are updated slowly during neural network training, the data-dependent LSH hash tables also changes slowly (this behavior was detailed in [9]). The amount of updates to the hash table drops very fast along with training, empirically verified in Fig. 1b (left) in Section 5. Based on this insight, we only need to update a small fraction of the hash table during training. To identify this subset $B \in [n]$ of nodes, gradient signals can be used. Intuitively, a node representation vector hashed into the wrong bucket will be aggregated with distant vectors and lead to larger errors and subsequently larger gradient signals. Specifically, we propose finding the candidate set $B$ of nodes by taking the union of the several buckets with the largest gradients, i.e., $B = \{i \mid h^{(l,k)}(i) = \arg\max_j [S_X^{(l,k)}]_{j,:}$ for some $k\}$. The memory and overhead required to update the entries in $B$ in the hash table is $O(|B|)$.

*(2) Update of LSH function:* In order to update the projection matrix $P$ that defines a SimHash $H^{(k)} : \mathbb{R}^d \to [c]$ (Eq. (3)), instead of the $O(n)$ full triplet loss introduced by [9], we consider a sampled version of the triplet loss on the candidate set $B$ with $O(|B|)$ complexity, namely

$$\mathcal{L}(H, \mathcal{P}_+, \mathcal{P}_-) = \max \left\{ 0, \sum_{(\boldsymbol{u},\boldsymbol{v}) \in \mathcal{P}_-} \cos(H(\boldsymbol{u}), H(\boldsymbol{v})) - \sum_{(\boldsymbol{u},\boldsymbol{v}) \in \mathcal{P}_+} \cos(H(\boldsymbol{u}), H(\boldsymbol{v})) + \alpha \right\}, \tag{7}$$

where $\mathcal{P}_+ = \{(\widetilde{X}_{i,:}, \widetilde{X}_{j,:}) \mid i,j \in B, \langle \widetilde{X}_{i,:}, \widetilde{X}_{j,:} \rangle > t_+\}$ and $\mathcal{P}_- = \{(\widetilde{X}_{:,i}, \widetilde{X}_{:,j}) \mid i,j \in B, \langle \widetilde{X}_{:,i}, \widetilde{X}_{:,j} \rangle < t_-\}$ are the similar and dissimilar node-pairs in the subset $B$; $t_+ > t_-$ and $\alpha > 0$ are hyper-parameters. This triplet loss $\mathcal{L}(H, \mathcal{P}_+, \mathcal{P}_-)$ is used to update $P$ using gradient descent, as described in [9], with a $O(c|B|d + |B|^2)$ overhead. Experimental validation of this LSH update mechanism can be found in Fig. 1b in Section 5.

**Avoiding $O(n)$ in Loss Evaluation.** We can estimate the final layer representation using the $r$ sketches $\{S^{(L,k)}\}_{k=1}^r$, i.e., $\widetilde{X}^{(L)} = \mathsf{Med}\{R^{(k)}S_X^{(L,k)} \mid k = 1,\cdots,r\}$ and compute the losses of all nodes for node classification (or some node pairs for link prediction). However, the complexity of loss evaluation is $O(n)$, proportional to the number of ground-truth labels. In order to avoid $O(n)$ complexity completely, rather than un-sketching the node representation for all labeled nodes, we employ the locality sensitive hashing (LSH) technique again for loss calculation so that only a subset of node losses are evaluated based on a set of hash tables. Specifically, we construct an LSH hash table for each class in a node classification problem, which indexes all of the labeled nodes of this class and can be utilized to choose the nodes with poor predictions by leveraging the locality property. This technique, introduced in [10], is known as sparse forward-pass and back-propagation, and we defer the descriptions to Appendix C.

**One-time Preprocessing.** If the convolution matrix $C$ is fixed (GCN, GraphSAGE), the "two-sided sketch" $S_C^{(l,k,k')} = \mathsf{CS}^{(k')}(\mathsf{TS}_k(C)^{\mathsf{T}}) \in \mathbb{R}^{c \times c}$ is the same in each layer and may be denoted as $S_C^{(k,k')}$. In addition, all of the $r^2$ sketches of $C$, i.e., $\{\{S_C^{(k,k')} \in \mathbb{R}^{c \times c}\}_{k=1}^r\}_{k'=1}^r$ can be computed during the preprocessing phase. If the convolution matrix $C$ is sparse (which is true for most GNNs following Eq. (1) on a sparse graph), we can use the sparse matrix representations for the sketches $\{\{S_C^{(k,k')} \in \mathbb{R}^{c \times c}\}_{k=1}^r\}_{k'=1}^r$, and the total memory taken by the $r^2$ sketches is $O(r^2 c(m/n))$ where $(2m/n)$ is the average node degree (see Appendix F for details). We also need to compute the $r$ count sketches of the input node feature matrix $X = X^{(0)}$, i.e., $\{S_X^{(0,k)}\}_{k=1}^r$ during preprocessing, which requires $O(rcd)$ memory in total. In this regard, we have substituted the input data with compact graph-size independent sketches (i.e., $O(c)$ memory). Although the preprocessing time required to compute these sketches is $O(n)$, it is a one-time cost prior to training, and it is widely known that sketching is practically very fast.

**Complexities of Sketch-GCN.** The theoretical complexities of Sketch-GNN is summarized as follows, where for simplicity we assume bounded maximum node degree, i.e., $m = O(n)$. **(1) Training Complexity**: *(1a) Forward and backward propagation*: $O(Lcrd(\log(c) + d + m/n)) = O(c)$ time and $O(Lr(cd + rm/n)) = O(c)$ memory. *(1b) Hash and sketch update*: $O(Lr(c + |B|d)) = O(c)$ time and memory. **(2) Preprocessing**: $O(r(rm + n + c)) = O(n)$ time and $O(rc(d + rm/n)) = O(c)$ memory. **(3) Inference**: $O(Ld(m + nd)) = O(n)$ time and $O(m + Ld(n + d)) = O(n)$ memory (the same as a standard GCN). We defer a detailed summary of the theoretical complexities of Sketch-GNN to Appendix F.

We generalize Sketch-GNN to more GNN models in Appendix D and the pseudo-code which outlines the complete workflow of Sketch-GNN can be find in Appendix E.

# 4  Related Work

**Towards sublinear GNNs.** Nearly all existing scalable methods focus on mini-batching the large graph and resolving the memory bottleneck of GNNs, without reducing the epoch training time. Few recent work focus on graph compression [22, 24] can also achieve sublinear training time by *coarsening* (e.g., using [31]) the graph during preprocessing or *condensing* the graph with dataset condensation techniques like gradient-matching [46], so that we can train GNNs on the coarsened/condensed graph with fewer nodes and edges. Nevertheless, these strategies suffer from **two issues**: **(1)** Although graph coarsening/condensation is a one-time cost, the memory and time overheads are often worse than $O(n)$ and can be prohibitively large on graphs with over 100K nodes. Even the fastest graph coarsening algorithm used by [22] takes more than 68 minutes to process the 233K-node *Reddit* graph [45]. The long preprocessing time renders any training speedups meaningless. **(2)** The test performance of a model trained on the coarsened graph highly depends on the GNN type. For graph condensation, if we do not carefully choose the GNN architecture used during condensation, the test performance of downstream GNNs can suffer from a 9.5% accuracy drop on the *Cora* graph [23]. For graph coarsening, although the performance of [22] on GCN is good, significant performance degradations are observed on GraphSAGE and GAT; see Section 5.

**Other scalable methods for GNNs** can be categorized into four classes, all of them still require linear training complexities. **(A)** On a large sparse graph with $n$ nodes and $m$ edges, the "full-graph" training of a $L$-layer GCN with $d$-dimensional (hidden) features per layer requires $O(m + ndL + d^2L)$ memory and $O(mdL + nd^2L)$ epoch time. **(B)** Sampling-based methods sample mini-batches from

the complete graph following three schemes: (1) node-wisely sample a subset of neighbors in each layer to reduce the neighborhood size; (2) layer-wisely sample a set of nodes independently in each layer; (3) subgraph-wisely sample a subgraph directly and simply forward-pass and back-propagate on that subgraph. **(B.1)** GraphSAGE [18] samples $r$ neighbors for each node while ignoring messages from other neighbors. $O(br^L)$ nodes are sampled in a mini-batch (where $b$ is the mini-batch size), and the epoch time is $O(ndr^L)$; therefore, GraphSAGE is impractical for deep GNNs on a large graph. FastGCN [12] and LADIES [48] are layer-sampling methods that apply importance sampling to reduce variance. **(B.2)** The subgraph-wise scheme has the best performance and is most prevalent. Cluster-GCN [14] partitions the graph into many densely connected subgraphs and samples a subset of subgraphs (with edges between subgraphs added back) for training per iteration. GraphSAINT [45] samples a set of nodes and uses the induced subgraph for mini-batch training. Both Cluster-GCN and GraphSAINT require $O(mdL + nd^2L)$ epoch time, which is the same as "full-graph" training, although Cluster-GCN also needs $O(m)$ pre-processing time. **(C)** Apart from sampling strategies, historical-embedding-based methods propose mitigating sampling errors and improving performance using some stored embeddings. GNNAutoScale [17] keeps a snapshot of all embeddings in CPU memory, leading to a large $O(ndL)$ memory overhead. VQ-GNN [15] maintains a vector quantized data structure for the historical embeddings, whose size is independent of $n$. **(D)** Linearized GNNs [42, 4, 33] replace the message passing operation in each layer with a one-time message passing during preprocessing. They are practically efficient, but the theoretical complexities remain $O(n)$. Linearized models usually over-simplify the corresponding GNN and limit its expressive power.

We defer discussion of more scalable GNN papers and the broad literature of sketching and LHS for neural networks to Appendix G.

## 5 Experiments

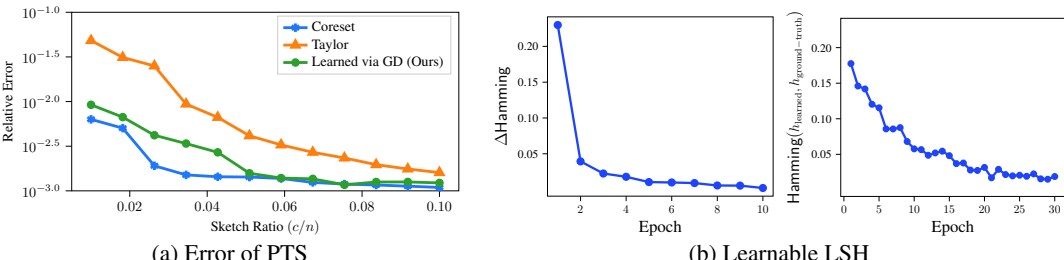

|   | (a) Error of PTS | (b) Learnable LSH |
|---|---|---|

Figure 1: **Figure 1a** Relative errors when applying polynomial tensor sketch (PTS) to the nonlinear unit $\sigma(CXW)$ following Eq. (4). The dataset used is Cora [34]. $\sigma$ is the sigmoid activation. We set $r = 5$ and test on a GCN with fixed $W = I_d \in \mathbb{R}^{d \times d}$. The coefficients $\{c_k\}_{k=1}^r$ can be computed by a coreset regression [19] (blue), by a Taylor expansion of $\sigma(\cdot)$ (orange), or learned from gradient descent proposed by us (green). **Figure 1b** The left plot shows the Hamming distance changes of the hash table in the 2nd layer during the training of a 2-layer *Sketch-GCN*, where the hash table is constructed from the unsketched representation $\widetilde{X}^{(1)}$ using SimHash. The right plot shows the Hamming distances between the hash table learned using our algorithm and the hash table constructed directly from $\widetilde{X}^{(1)}$.

Table 1: Performance of Sketch-GCN in comparison to Graph Condensation [23] and Graph Coarsening [22] on *Cora* and *Citeseer* with 2-layer GCNs.

| Benchmark | Cora | | Citeseer | |
|---|---|---|---|---|
| GNN Model | GCN | | | |
| "Full-Graph" (oracle) | .8119 ± .0023 | | .7191 ± .0018 | |
| Sketch-Ratio ($c/n$). | 0.013 | 0.026 | 0.009 | 0.018 |
| Coarsening | .3121 ± .0024 | .6518 ± .0051 | .5218 ± .0049 | .5908 ± .0045 |
| GCond | .7971 ± .0113 | .8002 ± .0075 | .7052 ± .0129 | .7059 ± .0087 |
| **Sketch-GNN (ours)** | .8012 ± .0104 | .8035 ± .0071 | .7091 ± .0093 | .7114 ± .0059 |

Table 2: Performance across GNN architectures in comparison to Graph Condensation [23] on *Cora* with sketch ratio $c/n = 0.026$.

| | Preprocessing Architecture | Downstream Architecture | |
|---|---|---|---|
| | | GCN | GraphSAGE |
| "Full-Graph" (oracle) | N/A | .8119 ± .0023 | .7981 ± .0053 |
| GCond | GCN | .7065 ± .0367 | .6024 ± .0203 |
| | GraphSAGE | .7694 ± .0051 | .7618 ± .0087 |
| Sketch-GNN (ours) | N/A | .8035 ± .0071 | .7914 ± .0121 |

In this section, we evaluate the proposed *Sketch-GNN* algorithm and compare it with the (oracle) "full-graph" training baseline, a graph-coarsening based approach (**Coarsening** [22]) and a dataset condensation based approach (**GCond** [23]) which enjoy sublinear training time, and other scalable

Table 3: Performance of Sketch-GNN in comparison to Graph Coarsening [22] on *ogbn-arxiv*.

| Benchmark | *ogbn-arxiv* | | | | | | | | |
|---|---|---|---|---|---|---|---|---|---|
| GNN Model | GCN | | | GraphSAGE | | | GAT | | |
| "Full-Graph" (oracle) | $.7174 \pm .0029$ | | | $.7149 \pm .0027$ | | | $.7233 \pm .0045$ | | |
| Sketch Ratio ($c/n$) | 0.1 | 0.2 | 0.4 | 0.1 | 0.2 | 0.4 | 0.1 | 0.2 | 0.4 |
| Coarsening | $.6508 \pm .0091$ | $.6665 \pm .0010$ | $.6892 \pm .0035$ | $.5264 \pm .0251$ | $.5996 \pm .0134$ | $.6609 \pm .0061$ | $.5177 \pm .0028$ | $.5946 \pm .0027$ | $.6307 \pm .0041$ |
| **Sketch-GNN (ours)** | $.6913 \pm .0154$ | $.7004 \pm .0096$ | $.7028 \pm .0087$ | $.6929 \pm .0194$ | $.6963 \pm .0056$ | $.7048 \pm .0080$ | $.6967 \pm .0067$ | $.6910 \pm .0135$ | $.7053 \pm .0034$ |

Table 4: Performance of Sketch-GNN versus SGC [42], GraphSAINT [45], and VQ-GNN [15].

| Benchmark | *ogbn-arxiv* | | | *Reddit* | | | *ogbn-product* | | |
|---|---|---|---|---|---|---|---|---|---|
| SGC | $.6944 \pm .0005$ | | | $.9464 \pm .0011$ | | | $.6683 \pm .0029$ | | |
| GNN Model | GCN | GraphSAGE | GAT | GCN | GraphSAGE | GAT | GCN | GraphSAGE | GAT |
| "Full-Graph" (oracle) | $.7174 \pm .0029$ | $.7149 \pm .0027$ | $.7233 \pm .0045$ | OOM | OOM | OOM | OOM | OOM | OOM |
| GraphSAINT | $.7079 \pm .0057$ | $.6987 \pm .0039$ | $.7117 \pm .0032$ | $.9225 \pm .0057$ | $.9581 \pm .0074$ | $.9431 \pm .0067$ | $.7602 \pm .0021$ | $.7908 \pm .0024$ | $.7971 \pm .0042$ |
| VQ-GNN | $.7055 \pm .0033$ | $.7028 \pm .0047$ | $.7043 \pm .0034$ | $.9399 \pm .0021$ | $.9449 \pm .0024$ | $.9438 \pm .0059$ | $.7524 \pm .0032$ | $.7809 \pm .0019$ | $.7823 \pm .0049$ |
| Sketch Ratio ($c/n$) | 0.4 | | | 0.3 | | | 0.2 | | |
| **Sketch-GNN (ours)** | $.7028 \pm .0087$ | $.7048 \pm .0080$ | $.7053 \pm .0034$ | $.9280 \pm .0034$ | $0.9485 \pm .0061$ | $.9326 \pm .0063$ | $.7553 \pm .0105$ | $.7762 \pm .0093$ | $.7748 \pm .0071$ |

methods including: a sampling-based method (**GraphSAINT** [45]), a historical-embedding based method (**VQ-GNN** [15]), and a linearized GNN (**SGC** [42]). We test on two small graph benchmarks including *Cora*, *Citeseer* and several large graph benchmarks including *ogbn-arxiv* (169K nodes, 1.2M edges), *Reddit* (233K nodes, 11.6M edges), and *ogbn-products* (2.4M nodes, 61.9M edges) from [20, 45]. See Appendix H for the implementation details.

**Proof-of-Concept Experiments: (1) Errors of gradient-learned PTS coefficients**: In Fig. 1a, we train the PTS coefficients to approximate the sigmoid-activated $\sigma(CXW)$ to evaluate its approximation power to the ground-truth activation. The relative errors are comparable to those of the coreset-based method. **(2) Slow-change phenomenon of LSH hash tables**: In Fig. 1b (left), we count the changes of the hash table constructed from an unsketched hidden representation for each epoch, characterized by the Hamming distances between consecutive updates. The changes drop rapidly as training progresses, indicating that apart from the beginning of training, the hash codes of most nodes do not change at each update. **(3) Sampled triplet loss for learnable LSH**: In Fig. 1b (right), we verify the effectiveness of our update mechanism for LSH hash functions as the learned hash table gradually approaches the "ground truth", i.e., the hash table constructed from the unsketched hidden representation.

**Performance of Sketch-GNNs.** We first compare the performance of *Sketch-GNN* with the other sublinear training methods, i.e., graph coarsening [22] and graph condensation [23] under various sketch ratios to understand how their performance is affected by the memory bottleneck. Since graph condensation (GCond) requires learning the condensed graph from scratch and cannot be scaled to large graphs with a large sketch ratio [23], we first compare with GCond and Coarsening on the two small graphs using a 2-layer GCN in Table 1. We see GCond and Sketch-GNN can outperform graph coarsening by a large margin and can roughly match the full-graph training performance. However, GCond suffers from a processing time that is longer than the training time (see below) and generalizes poorly across GNN architectures. In Table 2, we compare the performance of Sketch-GNN and GCond across two GNN architectures (GCN and GraphSAGE). While graph condensation (GCond) relies on a "reference architecture" during condensation, Sketch-GNN does not require preprocessing, and the sublinear complexity is granted by sketching "on the fly". In Table 2, we see the performance of GCond is significantly degraded when generalized across architectures, while Sketch-GNNs' performance is always close to that of full-graph training.

In Table 3, we report the test accuracy of both approaches on *ogbn-arxiv*, with a 3-layer GCN, GraphSAGE, or GAT as the backbone and a sketch ratio of 0.1, 0.2, or 0.4. We see there are significant performance degradations when applying Coarsening to GraphSAGE and GAT, even under sketch ratio 0.4, indicating that Coarsening may be compatible only with specific GNNs (GCN and APPNP as explained in [22]). In contrast, the performance drops of Sketch-GNN are always small across all architectures, even when the sketch ratio is 0.1. Therefore, our approach generalizes to more GNN architectures and consistently outperforms the Coarsening method.

We move on to compare Sketch-GNN with linearized GNNs (SGC), sampling-based (GraphSAINT), and historical-embedding-based (VQ-GNN) methods. In Table 4, we report the performance of

SGC, the "full-graph" training (oracle), GraphSAINT and VQ-GNN with mini-batch size 50K (their performance is not affected by the choice of mini-batch size if it is not too small), and Sketch-GNN with appropriate sketch ratios ($0.4$ on *ogbn-arxiv*, $0.3$ on *Reddit*, and $0.2$ on *ogbn-product*). From Table 4, we confirm that, with an appropriate sketch ratio, the performance of *Sketch-GNN* is usually close to the "full-graph" oracle and competitive with the other scalable approaches. The needed sketch ratio $c/n$ for Sketch-GNN to achieve competitive performance reduces as graph size grows. This further illustrates that, as previously indicated, the required training complexities (to get acceptable performance) are sublinear to the graph size.

**Efficiency of Sketch-GNNs.** For efficiency measures, we are interested in the comparison to Coarsening and GCond, since these two approaches achieve sublinear training time at the cost of some preprocessing overheads. Firstly, we want to address that both Coarsening and GCond suffer from an extremely long preprocessing time. On *ogbn-arxiv*, Coarsening and GCond require 358 and 494 seconds on average, respectively, to compress the original graph. In contrast, our Sketch-GNN sketch the input graph "on the fly" and does not suffer from a preprocessing overhead. On *ogbn-arxiv* with a learning rate of $0.001$, full-graph training of GCN for 300 epochs is more than enough for convergence, which only takes 96 seconds on average. The preprocessing time of Coarsening and GCond is much longer than the convergence time of full-graph training, which renders their training speedups meaningless. However, Sketch-GNN often requires more training memory than Coarsening and GCond to maintain the copies of sketches and additional data structures, although these memory overheads are small, e.g., only 16.6 MB more than Coarsening on *ogbn-arxiv* with sketch ratio $c/n = 0.1$. All three sublinear methods (Corasening, GCond, Sketch-GNN) lead to a denser adjacency/convolution matrix and thus increased memory per node. However, this overhead is small for Sketch-GNN because although we sketched the adjacency, its sparsity is still preserved to some extent, as sketching is a linear/multi-linear operation.

**Ablation Studies: (1) Dependence of sketch dimension $c$ on graph size $n$.** Although the theoretical approximation error increases under smaller sketch ratio $c/n$, we observe competitive experimental results with smaller $c/n$, especially on large graphs. In practice, the sketch ratio required to maintain "full-graph" model performance decreases with $n$. **(2) Learned Sketches versus Fixed Sketches.** We find that learned sketches can improve the performance of all models and on all datasets. Under sketch-ratio $c/n = 0.2$, the Sketch-GCN with learned sketches achieves $0.7004 \pm 0.0096$ accuracy on *ogbn-arxiv* while fixed randomized sketches degrade performance to $0.6649 \pm 0.0106$.

## 6    Conclusion

We present *Sketch-GNN*, a sketch-based GNN training framework with sublinear training time and memory complexities. Our main contributions are (1) approximating nonlinear operations in GNNs using polynomial tensor sketch (PTS) and (2) updating sketches using learnable locality-sensitive hashing (LSH). Our novel framework has the potential to be applied to other architectures and applications where the amount of data makes training even simple models impractical. The major limitation of Sketch-GNN is that the sketched nonlinear activations are less expressive than the original activation functions, and the accumulated error of sketching makes it challenging to sketch much deeper GNNs. We expect future research to tackle the above-mentioned issues and apply the proposed neural network sketching techniques to other types of data and neural networks. Considering broader impacts, we view our work mainly as a methodological and theoretical contribution, and there is no obviously foreseeable negative social impact.

## Acknowledgments and Disclosure of Funding

This work is supported by National Science Foundation NSF-IIS-FAI program, DOD-ONR-Office of Naval Research, DOD-DARPA-Defense Advanced Research Projects Agency Guaranteeing AI Robustness against Deception (GARD), Adobe, Capital One, JP Morgan faculty fellowships, and NSF DGE-1632976.

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
