# Supplementary Material for Sketch-GNN: Scalable Graph Neural Networks with Sublinear Training Complexity

## A  More Preliminaries

In this appendix, further preliminary information and relevant discussions are provided.

### A.1  Common GNNs in the Unified Framework

Here we list the common GNNs that can be re-formulated into the unified framework, which is introduced in Section 2. The majority of GNNs can be interpreted as performing message passing on node features, followed by feature transformation and an activation function, a process known as "generalized graph convolution" (Eq. (1)). Within this common framework, different types of GNNs differ from each other by their choice of convolution matrices $C^{(q)}$, which can be either fixed or learnable. A learnable convolution matrix depends on the inputs and learnable parameters and can be different in each layer (thus denoted as $C^{(l,q)}$),

$$C_{i,j}^{(l,q)} = \underbrace{\mathfrak{C}_{i,j}^{(q)}}_{\text{fixed}} \cdot \underbrace{h_{\theta^{(l,q)}}^{(q)}(X_{i,:}^{(l)}, X_{j,:}^{(l)})}_{\text{learnable}}, \tag{8}$$

where $\mathfrak{C}^{(q)}$ denotes the fixed mask of the $q$-th learnable convolution, which may depend on the adjacency matrix $A$ and input edge features $E_{i,j}$. While $h^{(q)}(\cdot, \cdot) : \mathbb{R}^{f_l} \times \mathbb{R}^{f_l} \to \mathbb{R}$ can be any learnable model parametrized by $\theta^{(l,q)}$. Sometimes a learnable convolution matrix may be further row-wise normalized as $C_{i,j}^{(l,q)} \leftarrow C_{i,j}^{(l,q)} / \sum_j C_{i,j}^{(l,q)}$, for example Graph Attention Network (GAT [39]). According to [15], we list some well-known GNN models that fall inside this framework in Table 5.

Table 5: Summary of GNNs re-formulated as generalized graph convolution [15].

| Model Name | Design Idea | Conv. Matrix Type | # of Conv. | Convolution Matrix |
|---|---|---|---|---|
| GCN[1] [27] | Spatial Conv. | Fixed | 1 | $C = \widetilde{D}^{-1/2} \widetilde{A} \widetilde{D}^{-1/2}$ |
| GIN[1] [44] | WL-Test | Fixed + Learnable | 2 | $\begin{cases} C^{(1)} = A \\ \mathfrak{C}^{(2)} = I_n \text{ and } h_{\epsilon^{(l)}}^{(2)} = 1 + \epsilon^{(l)} \end{cases}$ |
| SAGE[2] [18] | Message Passing | Fixed | 2 | $\begin{cases} C^{(1)} = I_n \\ C^{(2)} = D^{-1}A \end{cases}$ |
| GAT[3] [39] | Self-Attention | Learnable | # of heads | $\begin{cases} \mathfrak{C}^{(q)} = A + I_n \text{ and} \\ h_{\boldsymbol{a}^{(l,q)}}^{(q)}(X_{i,:}^{(l)}, X_{j,:}^{(l)}) = \exp\big(\text{LeakyReLU}( \\ \quad (X_{i,:}^{(l)}W^{(l,q)} \parallel X_{j,:}^{(l)}W^{(l,q)}) \cdot \boldsymbol{a}^{(l,q)})\big) \end{cases}$ |

[1] Where $\widetilde{A} = A + I_n$, $\widetilde{D} = D + I_n$.  [2] $C^{(2)}$ represents mean aggregator. Weight matrix in [18] is $W^{(l)} = W^{(l,1)} \parallel W^{(l,2)}$.
[3] Need row-wise normalization. $C_{i,j}^{(l,q)}$ is non-zero if and only if $A_{i,j} = 1$, thus GAT follows direct-neighbor aggregation.

### A.2  Definition of Locality Sensitivity Hashing

The definitions of count sketch and tensor sketch are based on the hash table(s) that merely require data-independent uniformity, i.e., a high likelihood that the hash-buckets are of comparable size. In contrast, locality sensitive hashing (LSH) is a hashing scheme with a locality-sensitive hash function $H : \mathbb{R}^d \to [c]$ that assures close vectors are hashed into the same bucket with a high probability while distant ones are not. Consider a locality-sensitive hash function $H : \mathbb{R}^d \to [c]$ that maps vectors in $\mathbb{R}^d$ to the buckets $\{1, \ldots, c\}$. A family of LSH functions $\mathcal{H}$ is $(D, tD, p_1, p_2)$-sensitive if and only if for any $\boldsymbol{u}, \boldsymbol{v} \in \mathbb{R}^d$ and any $H$ selected uniformly at random from $\mathcal{H}$, it satisfies

$$\begin{aligned} \text{if} \quad & \text{Sim}(\boldsymbol{u}, \boldsymbol{v}) \geq D \quad \text{then} \quad \mathbb{P}[H(\boldsymbol{u}) = H(\boldsymbol{v})] \geq p_1, \\ \text{if} \quad & \text{Sim}(\boldsymbol{u}, \boldsymbol{v}) \leq tD \quad \text{then} \quad \mathbb{P}[H(\boldsymbol{u}) = H(\boldsymbol{v})] \leq p_2, \end{aligned} \tag{9}$$

where $\text{Sim}(\cdot, \cdot)$ is a similarity metric defined on $\mathbb{R}^d$.

# B  Polynomial Tensor Sketch and Error Bounds

In this appendix, we provide additional theoretical details regarding the concentration guarantees of sketching the linear part in each GNN layer (Lemma 1), and the proof of our multi-layer error bound (Theorem 1).

## B.1  Error Bound for Sketching Linear Products

Here, we discuss the problem of approximating the linear product $CX^{(l)}W^{(l)}$ using count/tensor sketch. Since we rely on count/tensor sketch to compress the individual components $C$ and $X^{(l)}W^{(l)}$ of the intermediate product $CX^{(l)}W^{(l)}$ before we sketch the nonlinear activation, it is useful to know how closely sketching approximates the product. We have the following result:

**Lemma 1.** *Given matrices $C \in \mathbb{R}^{n \times n}$ and $(X^{(l)}W^{(l)})^{\mathsf{T}} \in \mathbb{R}^{d \times n}$, consider a randomly selected cuont sketch matrix $R \in \mathbb{R}^{c \times n}$ (defined in Section 2), where $c$ is the sketch dimension, and it is formed using $r = \sqrt[j]{n}$ underlying hash functions drawn from a 3-wise independent hash family $\mathcal{H}$ for some $j \geq 1$. If $c \geq (2 + 3^j)/(\varepsilon^2 \delta)$, we have*

$$\Pr\left(\left\|(CR_k^{\mathsf{T}})(R_k X^{(l)}W^{(l)}) - CX^{(l)}W^{(l)}\right\|_F^2 > \varepsilon^2 \|C\|_F^2 \|X^{(l)}W^{(l)}\|_F^2\right) \leq \delta. \quad (10)$$

*Proof.* The proof follows immediately from the Theorem 1 of [2]. $\qquad\square$

For $j \geq 1$ fulfilling $c \geq (2 + 3^j)/(\varepsilon^2 \delta)$, we have $j = O(\log_3 c)$, and consequently $r = (n)^{1/j} = \Omega(3^{\log_c n})$. In practice, when $n$ is not too small, $\log_c n \approx 1$ since $c$ grows sublinearly with respect to $n$. In this sense, the dependence of $r$ on $n$ is negligible.

## B.2  Proof of Error Bound for Final-Layer Representation (Theorem 1).

*Proof.* For fixed degree $r$ of a polynomial tensor sketch (PTS), by the Theorem 5 of [19], for $\Gamma^{(1)} = \max\left\{5\|X^{(l)}W^{(l)}\|_F^2, (2+3^r)\sum_i(\sum_j[X^{(l)}W^{(l)}]_{i,j})^r\right\}$, it holds that

$$\mathbf{E}(\|\sigma(CX^{(l)}W^{(l)}) - \widetilde{X}^{(l+1)}\|_F^2) \leq \left(\frac{2}{1 + \frac{c\lambda^{(l)2}}{nr\Gamma^{(l)}}}\right)\|\sigma(CX^{(l)}W^{(l)})\|_F^2, \quad (11)$$

where $\lambda^{(1)} \geq 0$ is the smallest singular value of the matrix $Z \in \mathbb{R}^{nd \times r}$, each column, $Z_{:,k}$, being the vectorization of $(CX^{(1)}W^{(1)})^{\odot k}$. This is the error bound for sketching a single layer, including the non-linear activation units.

Consider starting from the first layer ($l = 1$), for simplicity, let us denote the upper bound when $l = 1$ as $E_1$. The error in the second layer ($l - 2$), including the propagated error from the first layer $E_1$, is expressible as $\|\sigma(CX^{(2)}W^{(2)}) - \widetilde{X}^{(3)} + E_1\|_F^2$, which by sub-multiplicativity and the inequality $(a + b)^2 \leq 2a^2 + 2b^2$ gives

$$\|\sigma(CX^{(2)}W^{(2)}) - \widetilde{X}^{(3)} + E_1\|_F^2 \leq 2\|\sigma(CX^{(2)}W^{(2)}) - \widetilde{X}^{(3)}\|_F^2 + 2\|E_1\|. \quad (12)$$

By repeatedly invoking the update rule/recurrence in Eq. (1) and the Theorem 5 in [19] up to the final layer $l = L$, we obtain the overall upper bound on the total error as claimed. $\qquad\square$

# C  Learnable Sketches and LSH

## C.1  Learning of the Polynomial Tensor Sketch Coefficients.

We propose to learn the coefficients $\{c_k\}_{k=1}^r$ using gradient descent with an $L_2$ regularization, $\lambda \sum_{k=1}^r c_k^2$. For a node classification task, the coefficients in all layers are directly optimized to minimize the classification loss. Experimentally, the coefficients that obtain the best classification accuracy do not necessarily correspond to a known activation.

For the proof of concept experiment (Fig. 1a in Section 5), the coefficients $\{c_k\}_{k=1}^r$ in the first layer are learned to approximate the sigmoid activated hidden embeddings $\sigma(CX^{(1)}W^{(1)})$. The relative

errors are evaluated relative to the "sigmoid activated ground-truth". We find in our experiments that the relative errors are comparable to the coreset-based approach.

## C.2 Change the Hash Table of Count Sketches

Here we provide more information regarding the solution to the challenge (2) in Section 3.3. Since the hash tables utilized by each layer is different, we have to change the underlying hash table of the sketched representations when propagating through Sketch-GNN.

Consider the Sketch-GNN forward-pass described by Eq. (5), while the count sketch functions are now different in each layer. We denote the $k'$-th count sketch function in the $l$-th layer by $\mathsf{CS}^{(l,k')}(\cdot)$ (adding the superscript $(l)$), and denote its underlying hash table by $h^{(l,k)}$. Since the hash table used to count sketch $S_C^{(l,k,k')}$ is $h^{(l,k')}$, what we obtain using Eq. (5) is $\mathsf{CS}^{(l,k')}((X^{(l+1)})^\mathsf{T})$. However, we actually need $S_X^{(l+1,k')} = \mathsf{CS}^{(l+1,k')}((X^{(l+1)})^\mathsf{T})$ as the input to the subsequent layer.

By definition, we can change the underlying hash table like $S_X^{(l+1,k')} = \mathsf{CS}^{(l+1,k')}((X^{(l+1)})^\mathsf{T}) = \mathsf{CS}^{(l,k')}((X^{(l+1)})^\mathsf{T})R^{(l,k')}(R^{(l+1,k')})^\mathsf{T}$, where $R^{(l,k')}$ is the count sketch matrix of $\mathsf{CS}^{(l,k')}(\cdot)$. In fact, we only need to right multiply a $c \times c$ matrix $T^{(l,k')} := R^{(l,k')}(R^{(l+1,k')})^\mathsf{T}$, which is $O(c^2)$ and can be efficiently computed by

$$[T^{(l,k')}]_{i,j} = \sum_{a=1}^n s_a^{(l+1,k')} s_a^{(l,k')} \mathbb{1}\{h^{(l,k')}(a) = i\}\mathbb{1}\{h^{(l+1,k')}(a) = j\}. \tag{13}$$

We can maintain this $c \times c$ matrix $T^{(l,k')}$ as a signature of both hash tables $h^{(l,k')}$ and $h^{(l+1,k')}$. We are able to update $T^{(l,k')}$ efficiently when we update the hash tables on a subset $B$ of entries (see Section 3.3). We can also compute the sizes of buckets for both hash functions from $T^{(l,k')}$, which is useful to sketch the attention units in GAT; see Appendix D.

## C.3 Sparse Forward-Pass and Back-Propagation for Loss Evaluation

Here we provide more details on using the sparse forward-pass and back-propagation technique in [10] to avoid $O(n)$ complexity in loss evaluation. For a node classification task, we construct an LSH hash table for each class, which indexes all the labeled nodes in the training split that belong to this class. These LSH hash tables can be used to select the nodes with bad predictions in constant time, i.e., nodes whose predicted class scores have a small inner product with respect to the ground truth (one-hot encoded) label. Consequently, we only evaluate the loss on the selected nodes, avoiding the $O(n)$ complexity. The LSH hash tables are updated using the same method described in challenge (1) in Section 3.3.

# D    Generalize to More GNNs

This appendix briefly describes how to generalizing Sketch-GNN from GCN to some other GNN architectures, including GraphSAGE [18] and GAT [39].

## D.1    Sketch-GraphSAGE: Sketching Multiple Fixed Convolutions

The update rule (Eq. (5)) of Sketch-GNN can be directly applied to GNNs with only one fixed convolution matrix, such as GCN by setting $C = \widetilde{D}^{-1/2}\widetilde{A}\widetilde{D}^{-1/2}$. Here we seek to generalize Sketch-GNN to GNNs with multiple fixed convolutions, for example, GraphSAGE with $C^{(1)} = I_n$ and $C^{(2)} = D^{-1}A$. This can be accomplished by rewriting the update rule of GraphSAGE $X^{(l+1)} = \sigma(X^{(l)}W^{(l,1)} + D^{-1}AX^{(l)}W^{(l,2)})$ as a form resembling $\sigma(UV^\mathsf{T})$, so that the polynomial tensor sketch technique may still be used.

Therefore, we replace the update rule (Eq. (5)) with the following for GraphSAGE,

$$\sigma(X^{(l)}W^{(l,1)} + D^{-1}AX^{(l)}W^{(l,2)}) = \sigma\left(\left[I_n \parallel (D^{-1}A)^\mathsf{T}\right]^\mathsf{T}\left[X^{(l)}W^{(l,1)} \parallel X^{(l)}W^{(l,2)}\right]\right)$$

$$\approx \sum_{k=1}^{r} c_k \mathsf{TS}_k\left(\left[I_n\|(D^{-1}A)^\mathsf{T}\right]^\mathsf{T}\right)\mathsf{TS}_k\left(\left[X^{(l)}W^{(l,1)}\|X^{(l)}W^{(l,2)}\right]^\mathsf{T}\right)^\mathsf{T}. \tag{14}$$

## D.2  Sketch-GAT: Sketching Self-Attention Units

GAT employs self-attention to learn the convolution matrix $C^{(l)}$ (superscript $(l)$ denotes the convolution matrices learned are different in each layer). For the sake of simplicity, we assume single-headed attention while we can generalize to multiple heads using the same method as for GraphSAGE. The convolution matrix of GAT is defined as $C^{(l)} = (A + I_n) \odot ((\exp^{\odot}(Z^{(l)})\mathbf{1}_n)^\mathsf{T})^{-1} \exp^{\odot}(Z^{(l)})$, where $\mathbf{1}_n \in \mathbb{R}^n$ is a vector of ones, $Z^{(l)} \in \mathbb{R}^{n \times n}$ is the raw attention scores in the $l$-th layer, defined as $Z_{i,j}^{(l)} = \text{LeakyReLU}([X_{i,:}^{(l)}W^{(l)} \parallel X_{j,:}^{(l)}W^{(l)}] \cdot \boldsymbol{a}^{(l)})$, with $\boldsymbol{a}^{(l)} \in \mathbb{R}^{2n}$ being the learnable parameter vector.

Our goal is to approximate the sketches of the convolution matrix $S_C^{(l,k,k')}$ using the sketches of node representations $S_X^{(l,k)}$ and the learnable weights $W^{(l)}, \boldsymbol{a}^{(l)}$. We accomplish this by utilizing the locality-sensitive property of the sketches and by assuming that the random Rademacher variables $s^{(l,1)}, \cdots, s^{(l,k)}$ are fixed to $+1$. We find that setting all Rademacher variables to $+1$ has no discernible effect on the performance of Sketch-GAT.

With this additional assumption, each vector of node representation can be approximated by the average of vectors hashed into the same bucket, i.e., $X_{i,:}^{(l)} \approx \sum_j \mathbb{1}\{h^{(l,k)}(i) = h^{(l,k)}(j)\}X_{j,:}^{(l)} / \sum_j \mathbb{1}\{h^{(l,k)}(i) = h^{(l,k)}(j)\}$ for any $k \in [r]$. More specifically, the numerator is exactly the $h^{(l,k)}(i)$-th column vector of the sketch $S_X^{(l,k)}$, i.e., $\sum_j \mathbb{1}\{h^{(l,k)}(i) = h^{(l,k)}(j)\}X_{j,:}^{(l)} = [S_X^{(l,k)}]_{:,h^{(l,k)}(i)}$. Using only the sketch $S_X^{(l,k)}$ and the bucket sizes in the hash table $h^{(l,k)}$, we can approximate any $X_{i,:}^{(l)}$ as a function of $h^{(l,k)}(i)$ (instead of $i$), and thus approximate the entries of this $n \times n$ matrix $Z^{(l)}$ with $c^2$ distinct values only. Even after the element-wise exponential and row-wise normalization, any attention score $[((\exp^{\odot}(Z^{(l)})\mathbf{1}_n)^\mathsf{T})^{-1} \exp^{\odot}(Z^{(l)})]_{i,j}$ can still be estimated as a function of the tuple $(h^{(l,k)}(i), h^{(l,k)}(j))$, where $Z_{i,j}^{(l)} = \langle X_{i,:}^{(l)}, X_{j,:}^{(l)}\rangle$. This means we can approximate the attention scores $[((\exp^{\odot}(Z^{(l)})\mathbf{1}_n)^\mathsf{T})^{-1} \exp^{\odot}(Z^{(l)})]$ using the sketched representation $S_X^{(l,k)}$, using the fact that $Z_{i,j}^{(l)} = \langle X_{i,:}^{(l)}, X_{j,:}^{(l)}\rangle \approx \langle [S_X^{(l)}]_{:,h^{(l)}(i)}/|\{a|h^{(l)}(a) = h^{(l)}(i)\}|[S_X^{(l)}]_{:,h^{(l)}(j)}/|\{a|h^{(l)}(a) = h^{(l)}(j)\}|\rangle$, where $\{a|h^{(l)}(a) = h^{(l)}(i)\}$ is the bucket size of $h^{(l)}(i)$-th hash bucket.

We can see that computing the sketches of $C^{(l)}$ (the sketch functions are defined by the same hash table $h^{(l,k)}(\cdot)$) only requires **(1)** the $c^2$ distinct estimations of the entries in $((\exp^{\odot}(Z^{(l)})\mathbf{1}_n)^\mathsf{T})^{-1} \exp^{\odot}(Z^{(l)})$, and **(2)** an "averaged $c \times c$ version" of the mask $(A + I_n)$, which is exactly the two-sided count sketch of $(A + I_n)$ defined by the hash table $h^{(i,j)}$. In conclusion, we find a $O(c^2)$ algorithm to estimate the sketches of the convolution matrix $S_C^{(l,k,k')}$ using the sketches of node representations $S_X^{(l,k)}$ and a pre-computed two-sided count sketch of the mask matrix $(A + I_n)$.

## E  The Complete Pseudo-Code

The following is the pseudo-code outlining the workflow of Sketch-GNN (assuming GCN backbone).

## F  Summary of Theoretical Complexities

In this appendix, we provide more details on the theoretical complexities of Sketch-GNN with a GCN backbone. For simplicity, we assume bounded maximum node degree, i.e., $m = \theta(n)$.

**Algorithm 1** *Sketch-GNN*: sketch-based approximate training of GNNs with sublinear complexities

---

**Require:** GNN's convolution matrix $C$, input node features $X$, ground-truth labels $Y$
1  **procedure** PREPROCESS($C, X$)
2     |  Sketch $X = X^{(0)}$ into $\{S_X^{(0,k)}\}_{k=1}^r$ and sketch $C$ into $\{\{S_C^{(k,k')}\}_{k=1}^r\}_{k'=1}^r$
3  **procedure** TRAIN($\{\{S_C^{(k,k')}\}_{k=1}^r\}_{k'=1}^r, \{S_X^{(0,k)}\}_{k=1}^r, Y$)
4     |  Initialize weights $\{W^{(l)}\}_{l=1}^L$, coefficients $\{\{c_k^{(l)}\}_{k=1}^r\}_{l=1}^L$, and LSH projections $\{\{P_k^{(l)}\}_{k=1}^r\}_{l=1}^L$.
5     |  **for** epoch $t = 1, \ldots, T$ **do**
6     |     |  **for** layer $l = 1, \ldots, L - 1$ **do**
7     |     |     |  Forward-pass and compute $S_X^{(l+1,k')}$ via Eq. (5).
8     |     |  Evaluate losses on a subset $B$ of nodes in buckets with the largest gradients for each class.
9     |     |  Back-propagate and update weights $\{W^{(l)}\}_{l=1}^L$ and coefficients $\{\{c_k^{(l)}\}_{k=1}^r\}_{l=1}^L$.
10     |     |  Update the LSH projections $\{\{P_k^{(l)}\}_{k=1}^r\}_{l=1}^L$ with the triplet loss Eq. (7) for every $T_{\text{LSH}}$ epoch.
11     |  **return** Learned weights $\{W^{(l)}\}_{l=1}^L$
12 **procedure** INFERENCE($\{W^{(l)}\}_{l=1}^L$)
13     |  Predict via the corresponding standard GNN update rule, using the learned weights $\{W^{(l)}\}_{l=1}^L$

---

**Preprocessing.** The $r$ sketches of the node feature matrix take $O(r(n + c)d)$ time and occupy $O(rdc)$ memory. And the $r^2$ sketches of the convolution matrix require $O(r(m + c) + r^2 m)$ time (the LSH hash tables are determined by the node feature vectors already) and $O(r^2 cm/n)$ memory. The total preprocessing time is $O(r^2 m + rm + r(n + c)d) = O(n)$ and the memory taken by the sketches is $O(rc(d + rm/n)) = O(c)$.

**Forward and backward passes.** For each sketch in each layer, matrix multiplications take $O(cd(d + m/n))$ time, FFT and its inverse take $O(dc \log(c))$ time, thus the total forward/backward pass time is $O(Lcrd(\log(c) + d + m/n)) = O(c)$. The memory taken by sketches in a Sketch-GNN is just $L$ times the memory of input sketches, i.e., $O(Lrc(d + rm/n)) = O(c)$.

**LSH hash updates and loss evaluation.** Computing the triplet loss and updating the corresponding part of the hash table requires $O(Lrb(n/c))$ where $b = |B|$ is the number of nodes selected based on the gradients (for each sketch). Updates of the sketches are only performed every $T_{\text{LSH}}$ epochs.

**Inference** is conducted on the standard GCN model with parameters $\{W^{(l)}\}_{l=1}^L$ learned via Sketch-GNN, which takes $O(Ld(m/n + d))$ time on average for a node sample.

**Remarks.** **(1) Sparsity of sketched convolution matrix.** The two-sided sketch $\mathsf{CS}(\mathsf{CS}(C)^{\mathsf{T}}) \in \mathbb{R}^{c \times c}$ maintains sparsity for sparse convolution $C$, as $\mathsf{CS}(\mathsf{CS}(C)^{\mathsf{T}}) = RCR^{\mathsf{T}}$ (a product of 3 sparse matrices) is still sparse, where count-sketch matrix $R \in \mathbb{R}^{c \times n}$ has one non-zero entry per column (by its definition see Section 2). If $C$ has at most $s$ non-zeros per column, there are $\leq s$ non-zeros per column in $RC$ when $c \gg s$ (holds for sparse graphs that real-world data exhibits). Thus, we avoid the $O(c^2)$ memory cost and are strictly $O(c)$. **(2) Overhead of computing the LSH hash tables.** Following Eq. (3) and Eq. (6), we need $O(cd)$ overhead to obtain the LSH hash index of each node, and since we have $n$ nodes in total and we maintain $r$ independent hash tables, the total overhead for computing the LSH hash tables is $O(ncrd)$ during preprocessing.

In conclusion, we achieve sublinear training complexity except for the one-time preprocessing step.

# G   More Related Work Discussions

## G.1   Sketch-GNN v.s. GraphSAINT

GraphSAINT[45] is a graph sampling method that enables training on a mini-batch of subgraphs instead of on the large input graph. GraphSAINT is easily applicable to any graph neural network (GNN), introduces minor overheads, and usually works well in practice. However, GraphSAINT is not a sub-linear training algorithm, it saves memory at the cost of time overhead. We have to iterate through the full batch of subgraphs in an epoch, and the training time complexity is still linear in the graph size. In contrast, our proposed Sketch-GNN is an approximated training algorithm of some GNNs with sub-linear time and memory complexities. Sketch-GNN has the potential to scale better than GraphSAINT on larger graphs. Besides, as a sketching algorithm, Sketch-GNN is suitable for

some scenarios, for example, sketching big graphs in an online/streaming fashion. Sketch-GNN can also be combined with subgraph sampling to scale up to extremely large graphs. Sketching the sampled subgraphs (instead of the original graph) avoids the decreasing sketch-ratio when the input graph size grows to extremely large while with a fixed memory constraint.

## G.2 Sketching in GNNs

EXACT [30] is a recent work which applies random projection to reduce the memory footprint of non-linear activations in GNNs. In this regard, they also applies sketching techniques to scale up the training of GNNs. However there are three important differences between Sketch-GNN and EXACT summarized as follows: (1) Sketch-GNN propagates sketched representations while sketching in EXACT only affects the back-propagation, (2) Sketch-GNN sketches the graph size dimension while EXACT sketches the feature dimension, and (3) Sketch-GNN enjoys sub-linear complexity while EXACT does not. We want to address that Sketch-GNN and EXACT are aiming for very different goals; Sketch-GNN is sketching the graph to achieve sub-linear complexity, while EXACT is sketching to save the memory footprint of non-linear activations

## G.3 Sketching Neural Networks

Compression of layers/kernels via sketching methods has been discussed previously, but not on a full-architectural scale. Wang et al. [40] utilize a multi-dimensional count sketch to accelerate the decomposition of a tensorial kernel, at which point the tensor is fully-restored, which is not possible in our memory-limited scenario. Shi and Anandkumar [35] utilize the method of Wang et al. [40] to compute compressed tensorial operations, such as contractions and convolutions, which is more applicable to our setup. Their experiments involve the replacement of a fully-connected layer at the end of a tensor regression network rather than full architectural compression. Furthermore, they guarantee the recovery of a sketched tensor rather than the recovery of tensors passing through a nonlinearity such as a ReLU. Kasiviswanathan et al. [26] propose layer-to-layer compression via sign sketches, albeit with no guarantees, and their back-propagation equations require $O(n^2)$ memory complexity when dealing with the nonlinear activations. In contrast to these prior works, we propose a sketching method for nonlinear activation units, which avoids the need to unsketch back to the high dimensional representation in each layer.

## G.4 LSH in Neural Networks

Locality sensitive hashing (LSH) has been widely adopted to address the time and memory bottlenecks of many large-scale neural networks training systems, with applications in computer vision [13], natural language processing [6] and recommender systems [36]. For fully connected neural networks, Chen et al. [10] proposes an algorithm, SLIDE, that retrieves the neurons in each layer with the maximum inner product during the forward pass using an LSH-based data structure. In SLIDE, gradients are only computed for neurons with estimated large gradients during back-propagation. For transformers, Kitaev et al. [28] proposes to mitigate the memory bottleneck of self-attention layers over long sequences using LSH. More recently, Chen et al. [9] has dealt with the update overheads of LSH during the training of NNs. Chen et al. [9] introduces a scheduling algorithm to adaptively perform LSH updates with provable guarantees and a learnable LSH algorithm to improve the query efficiency.

## G.5 Graph Sparsification for GNNs

Graph sparsification, i.e., removing task-irrelevant and redundant edges from the large input graph, can be applied to speed up the training of GNNs. Calandriello et al. [5] propose fast and scalable graph sparsification algorithms for graph-Laplacian-based learning on large graphs. Zheng et al. [47] sparsify the graph using neural networks and applied to the training of general GNNs. Srinivasa et al. [38] specifically considered the graph sparsification problem for graph attention (e.g., graph attention networks, GAT). Graph sparsification will not reduce the number of nodes; thus, the memory reduction of node feature representation is limited. However, some carefully designed graph sparsification may enjoy small approximation error (thus smaller performance drops) and improve the robustness of learned models.

# H   Implementation Details

This appendix lists the implementation details and hyper-parameter setups for the experiments in Section 5.

**Datasets.** Dataset *ogbn-arxiv* and *ogbn-product* are obtained from the Open Graph Benchmark (OGB)[1]. Dataset *Reddit* is adopted from [45] and downloaded from the PyTorch Geometric library[2], it is a sparser version of the original dataset provided by Hamilton et al. [18]. We conform to the standard data splits defined by OGB or PyTorch Geometric.

**Code Frameworks.** The implementation of our **Sketch-GNN** is based on the PyTorch library and the PyTorch Sparse library[3]. More specifically, we implement the Fast Fourier Transform (FFT) and its inverse (used in tensor sketch) using PyTorch. We implement count sketch of node features and convolution matrices as sparse-dense or sparse-sparse matrix multiplications, respectively, using PyTorch Sparse. Our implementations of the standard GNNs are based on the PyTorch Geometric library. The implementations of SGC [42] and GraphSAINT [45] are also adopted from PyTorch Geometric, while the implementations of VQ-GNN[4] [15] and Coarsening[5] [22] are adopted from their official repositories, respectively. All of the above-mentioned libraries (except for PyTorch) and code repositories we used are licensed under the MIT license.

**Computational Infrastructures.** All of the experiments are conducted on Nvidia RTX 2080Ti GPUs with Xeon CPUs.

**Repeated Experiments.** For the efficiency measures in Section 5, the experiments are repeated two times to check the self-consistency. For the performance measures in Section 5, we run all the experiments five times and report the mean and variance.

**Setups of GNNs and Training.** On all of the three datasets, unless otherwise specified, we always train 3-layer GNNs with hidden dimensions set to 128 for all scalable methods and for the oracle "full-graph" baseline. The default learning rate is 0.001. We apply batch normalization on *ogbn-arxiv* but not the other two datasets. Dropout is never used. Adam is used as the default optimization algorithm.

**Setups of Baseline Methods.** For SGC, we set the number of propagation steps $k$ in preprocessing to 3 to be comparable to other 3-layer GNNs. For GraphSAINT, we use the GraphSAINT-RW variant with a random walk length of 3. For VQ-GNN, we set the number of K-means clusters to 256 and use a random walk sampler (walk length is also 3). For Coarsening, we use the Variation Neighborhood graph coarsening method if not otherwise specified. As reported in [22], this coarsening algorithm has the best performance. We use the mean aggregator in GraphSAGE and single-head attention in GAT.

**Setups of Sketch-GNN.** If not otherwise mentioned, we always set the polynomial order (i.e., the number of sketches) $r = 3$. An $L_2$ penalty on the learnable coefficients is applied with coefficient $\lambda$ ranging from 0.01 to 0.1. For the computation of the triplet loss, we always set $\alpha$ to 0.1, but the values of $t_+ > t_- > 0$ are different across datasets. We can find a suitable starting point to tune by finding the smallest inner product of vectors hashed into the same bucket. To get the sampled subset $B$, we take the union of $0.01c$ buckets with the largest gradient norms for each sketch. The LSH hash functions are updated every time for the first 5 epochs, and then only every $T_{\text{LSH}} = 10$ epochs. We do not traverse through all pairs of vectors in $B$ to populate $\mathcal{P}_+$ and $\mathcal{P}_-$. Instead, we randomly sample pairs until $|\mathcal{P}_+|, |\mathcal{P}_-| > 1000$.

---

[1] https://ogb.stanford.edu/
[2] https://github.com/pyg-team/pytorch_geometric
[3] https://github.com/rusty1s/pytorch_sparse
[4] https://github.com/devnkong/VQ-GNN
[5] https://github.com/szzhang17/Scaling-Up-Graph-Neural-Networks-Via-Graph-Coarsening