# OpenReview forum: "Sketch-GNN: Scalable Graph Neural Networks with Sublinear Training Complexity"
_NeurIPS.cc/2022/Conference — NeurIPS 2022 Accept_

### Official Review · Reviewer_HVXV · 2022-07-07

**Rating:** 6
**Confidence:** 4
**Soundness:** 4 excellent
**Presentation:** 4 excellent
**Contribution:** 3 good

**Summary:**

Authors tackle the problem of scalability of GNNs to very large graphs. They propose Sketch-GNN method that approximates computation of a typical message-passing GNN in time that is sublinear of the number of nodes. This is achieved by “compressing” the graph (i.e., its adjacency matrix, or generally a convolution matrix, and the node feature matrix) by using Count Sketch and devising an approximation to a GNN layer computation, that is built on Polynomial Tensor Sketch with Locality Sensitive Hashing. Their approach is applied to multiple GNN kernels: GCN, GAT, and GraphSAGE. On 3 datasets (ogbn-arxiv, ogbn-product, and Reddit) Sketch-GNN shows strong classification performance competitive with VQ-GNN, however GraphSAINT is comparable or better. In terms of run-time efficiency, Sketch-GNN significantly improved over compared models on the evaluated medium-sized datasets.

**Questions:**

1. Have you empirically tested the effect of choice of the number of sketches $r$ on the prediction performance and computation time & memory? Appears all experiments are done with $r=3$. For sketch ratio $c / n = 0.1$ the lower bound $3^{log_c n}$ is closer to 4 for all tested graph sizes.
2. Sketch-GNN with GAT kernel is more expensive than GCN as the convolution matrix is not shared across layers. To how much longer computation time and increased memory footprint does this translate?
3. For efficiency experiments presented in Table 1, authors used sketch ratio $c / n = 0.1$ which may be an optimistic scenario. What does the efficiency comparison looks like with settings as used in the prediction performance comparison in Table 3, where higher sketch ratios are used?
4. Is this approach applicable to link prediction tasks? Would that necessitate potentially expensive un-sketching of node representations?
5. As mentioned above under "weaknesses", could Sketch-GNN scale to datasets such as MAG240M-LSC? What graph size do you estimate as a practical limit on current hardware?
6. Have you considered a combination of subgraph sampling methods (e.g., GraphSAINT) and Sketch-GNN?

**Limitations:**

Method limitations are not explicitly discussed, nor any potential negative societal impact.

**Strengths And Weaknesses:**

### Strengths
- Motivation: This paper tackles an important problem, scaling GNNs to graphs of millions or billions of nodes has clear industrial applications.
- Clarity: The paper is very well written and easy to follow.
- Correctness: I have read through and believe the method derivation to be correct, however I do not have a deep expertise in sketch and LSH methodology.
- Originality: The proposed method achieves similar goal to VQ-GNN but by different means. It builds on existing Polynomial Tensor Sketch and Locality Sensitive Hashing, devising online learning of Sketches such that the whole stack of GNN layers is trainable without realizing explicit representation of all N nodes.

### Weaknesses
- Limitation: Authors do not perform experiments on MAG240M-LSC (contains ~240M nodes). I believe their approach would still not scale to graphs with hundreds of millions of nodes, as the sketch ratio would need to be very low, which may not be practical. I would like to see a discussion from the authors on this point.
- Achieved prediction performance is comparable or lower to GraphSAINT – a sampling-based approach; compared to VQ-GNN the prediction performance appears to be equivalent.

### Minor comments:
- A diagram illustration would further improve readability of the paper.
- Typos: line 131: “sorely” -> “solely”?; line 234: “ldots” command

In summary, the proposed method is a valuable contribution, albeit its practical limitations should be further discussed in the Experiments section and Conclusion.

---

> ### Author Response · Authors · 2022-08-02
> **$\newcommand{HVXV}{\textcolor{red}{\mathrm{HVXV}}}$Response to Reviewer $\HVXV$ (1/3)**
>
> $\newcommand{HVXV}{\textcolor{red}{\mathrm{HVXV}}}$
> We appreciate Reviewer $\HVXV$'s feedback and review. We are encouraged that Reviewer $\HVXV$ thinks we tackle an important problem and is a valuable contribution. Reviewer $\HVXV$ raises some questions about the practical limitations of our algorithm. We would like to address Reviewer $\HVXV$'s questions below.
>
> > Q1: **Practicality on MAG240M-LSC**: *Authors do not perform experiments on MAG240M-LSC (contains ~240M nodes). I believe their approach would still not scale to graphs with hundreds of millions of nodes, as the sketch ratio would need to be very low, which may not be practical. I would like to see a discussion from the authors on this point.*
>
> Thanks for this question. We agree it is an interesting question to discuss whether our sketch-GNN algorithm can scale to graphs with hundreds of millions of nodes (thousand times larger ogbn-arxiv) and even larger graphs. From Theorem 1 we know the relative error of the estimated final layer representation $\tilde{X}^{(L)}$ in Sketch-GCN is $O(L(n/c))$, i.e., inversely proportional to the sketch ratio $c/n$. Therefore, when scaling up the graph size $n$ with a fixed resource limit (e.g., fixed GPU memory capacity), the sketch ratio becomes smaller and smaller, and consequently, we cannot bound the estimation error. A possible way to get around this is to combine subgraph sampling with sketching so that we only need to sketch on subgraphs of constant sizes, no matter how large the input graph size $n$ is. This is related to the Q8 below, so we postpone relevant discussions there.
>
> > Q2: **Compare with GraphSAINT**: *Achieved prediction performance is comparable or lower to GraphSAINT – a sampling-based approach; compared to VQ-GNN the prediction performance appears to be equivalent.*
>
> * We want to address that GraphSAINT and VQ-GNN are mini-batch sampling-based methods. They do not compress the graph representation and still do a complete pass on the original graph when iterating through the subgraphs in the mini-batch. **The training time of GraphSAINT and VQ-GNN is still linear in the graph size**.
> * **Sketch-GNN is a sketching-based method that enjoys sub-linear training complexities at the cost of the approximation errors of sketching**. We think it is not very appropriate to treat GraphSAINT/VQ-GNN and Sketch-GNN as the same class of algorithms and just compare their performance since the efficiency boost they provided is different.
>
> > Q3: **Minor Comments**: *A diagram illustration would further improve the readability of the paper. Typos: line 131: “sorely” -> “solely”?; line 234: “ldots” command*
>
> Thank you for pointing out these typos, and we have already corrected them.
>
> > Q4: **Choice of r**: *Have you empirically tested the effect of choice of the number of sketches $r$ on the prediction performance and computation time & memory? Appears all experiments are done with $r=3$. For sketch ratio $c/n=0.1$ the lower bound $3^{\log_cn}$ is closer to 4 for all tested graph sizes.*
>
> Firstly we want to kindly note that from Lemma 1, we know $r=\Omega(3^{\log_cn})$, instead of saying $r$ should be set to $3^{\log_cn}$. It should be understood as follows: the dependence of $r$ on $n$ is negligible when $n$ is not too small, and we can treat it as a constant in practice (Line 200). In practice, we found setting $r\geq4$ may lead to increased instability during training which may harm the performance, and therefore we fix it to $r=3$.
>
> > Q5: **Sketch-GAT**: *Sketch-GNN with GAT kernel is more expensive than GCN as the convolution matrix is not shared across layers. To how much longer computation time and increased memory footprint does this translate?*
>
> For Sketch-GAT trained on ogbn-arxiv with one head, the pre-processing time is 38s, the epoch time is 0.42s, and the peak training memory is 92.7MB. For the GAT with the same architecture trained with GraphSAINT, the epoch time is 0.34s, and the peak training memory is 59.1MB. Although Sketch-GAT suffices from more expensive overhead to approximate the sketched attention units in each layer, Sketch-GNN enjoys sub-linear complexities while GraphSAINT does not.

---

> > ### Author Response · Authors · 2022-08-02
> > **$\newcommand{HVXV}{\textcolor{red}{\mathrm{HVXV}}}$Response to Reviewer $\HVXV$ (2/3)**
> >
> > > Q6: *For efficiency experiments presented in Table 1, the authors used sketch ratio $c/n=0.1$, which may be an optimistic scenario. What does the efficiency comparison looks like with settings as used in the prediction performance comparison in Table 3, where higher sketch ratios are used?*
> >
> > Thanks for your question. We want to highlight that for the efficiency comparison, setting the sketch ratio $c/n=0.1$ is not necessarily an optimistic scenario for Sketch-GNN (we didn't cherry-pick but just selected a random compression ratio to report). No matter what sketch ratio $c/n$ is used, the efficiency comparison to the baselines is still fair (the full-graph training case is not a baseline) because we made sure to keep the size of the graph fed into the GNN model to be the same for all baselines. Apart from the "full-graph training" case, there are two types of baselines: mini-batch sampling methods (GraphSAINT, VQ-GNN) and compression methods (graph coarsening). For sampling baselines, we have set the (average) subgraph size $b=0.1n$, and for compression methods, we have set the size of coarsened graph to be $0.1n$ too. Under this setup, the effective number of nodes fed to the model in sketch-GNN and the other baselines are all proportional to the sketch ratio $c/n$, no matter what sketch ratio we choose. If we increase $c/n$, the memory usage and epoch time of all methods will increase. We will post added efficiency evaluation results to the appendices in future updates.
> >
> > > Q7: **Link Prediction**: *Is this approach applicable to link prediction tasks? Would that necessitate potentially expensive un-sketching of node representations?*
> >
> > Thanks for your interesting question. If link prediction is made through a top model on a pair of node representations, then at least un-sketching the node representations of this pair of nodes is needed. However, the link prediction model is usually trained on some sampled edges per iteration instead of all of the edges at once. We think we do not have to un-sketch the full node representation in each iteration. Designing this sketch-compatible link prediction layer is non-trivial and out of the scope of this current paper, but we will try to add related discussions to the appendices in future updates.

---

> > > ### Author Response · Authors · 2022-08-02
> > > **$\newcommand{HVXV}{\textcolor{red}{\mathrm{HVXV}}}$Response to Reviewer $\HVXV$ (3/3)**
> > >
> > > $\newcommand{HVXV}{\textcolor{red}{\mathrm{HVXV}}}$
> > > > Q8: **Combination with Subgraph Sampling**: *As mentioned above under "weaknesses", could Sketch-GNN scale to datasets such as MAG240M-LSC? What graph size do you estimate as a practical limit on current hardware? Have you considered a combination of subgraph sampling methods (e.g., GraphSAINT) and Sketch-GNN?*
> > >
> > > We really appreciate your insightful idea. If we directly sketch the input graph, since the estimation error is inversely proportional to the sketch ratio $c/n$, and from Table 3, we can estimate that sketch ratio $c/n$ at around 0.1 should work for graphs much larger than ognb-product. We estimate the practical limit on a 12GB memory GPU would be around 20M nodes (if there are 128 features per node).
> > >
> > > To scale up to graphs with 240M nodes or larger, as discussed in answer to Q1 above, combining with subgraph sampling methods is one way to get around increasing estimation error (which is inversely proportional to the sketch ratio $c/n$) when the input graph size $n$ grows to extremely large while fixing the sketched size $c$ due to device constraints. However, we also want to address a few practical issues when scaling up the combined algorithm to graphs with hundreds of millions of nodes, e.g., MAG240M-LSC.
> > > 1. Since the sketched representation of different subgraphs is different, we have to pre-process each subgraph and keep the batch of sketched representations in CPU memory. Although the time complexity of the pre-processing process is still linear in $n$ and the memory of resulted sketches is still sub-linear, graphs like MAG240M-LSC (whose dumped file size is around 200G) can run out of CPU memory during pre-processing. A way to get around this is to load the dataset to CPU memory and pre-process in a streaming fashion, in parallel to the training process. However, implementing such an algorithm requires more effort.
> > > 2. If the GPU memory can only fit in one sketched representation of one subgraph, then to train an epoch, we have to repeatedly load the sketched representation of the subgraphs to GPU, which may lead to a large time overhead. A way to get around this is multi-GPU training but implementing such an algorithm requires more effort.
> > >
> > > We believe, in principle, the combination of subgraph sampling and sketch-GNN has the potential to scale to graphs like MAG240M-LSC. But given the practical reasons mentioned above, a lot of work is needed to implement the algorithm for the current hardware. Thus we may leave the empirical implementation to future work.
> > >
> > > On the other hand, in order to verify that combining subgraph sampling and sketching is feasible and does not lead to further performance drop, we implement it for sketch-GCN on ogbn-arxiv, with a $0.1n$ average subgraph size and 0.4 sketch ratio (so that the effective compress ratio is still 0.4, which is not affected by the sampling process). The resulted performance is $0.6939\pm0.0150$, within the error bar of the non-sampling sketch-GCN baseline $0.7028\pm0.0087$ with a 0.4 sketch ratio (Table 3). **This experiment shows the compatibility between subgraph sampling and Sketch-GNN and the feasibility of scaling up to larger graphs using the combined algorithm.**
> > >
> > > ---
> > >
> > > We again thank Reviewer $\HVXV$ for the time and effort in reviewing our paper. We hope that our explanations above have addressed all concerns of Reviewer $\HVXV$. If so, we would greatly appreciate it if Reviewer $\HVXV$ could consider raising their score. We are happy to answer any further questions.

---

> > > > ### Comment · Reviewer_HVXV · 2022-08-03
> > > > **Reply to authors**
> > > >
> > > > Thank you for the thorough answers to my questions!
> > > > Ultimately, because of the practical limitations as discussed here, that should be included in the main text if the paper is accepted, I keep the original rating 6. The definition of the rating 6 ("Technically solid, moderate-to-high impact paper, with no major concerns with respect to evaluation, resources, reproducibility, ethical considerations") best summarizes the paper in my opinion and I could not justify increasing of the score to 7.

---

### Official Review · Reviewer_wwt9 · 2022-07-08

**Rating:** 7
**Confidence:** 4
**Soundness:** 3 good
**Presentation:** 3 good
**Contribution:** 3 good

**Summary:**

Training GNNs on large graphs is a long standing challenge. This work proposes a sketch-based algorithm whose training time and memory grow sublinearly with respect to graph size by training. Specifically, the author provides a novel protocol for sketching non-linear activations and graph convolution matrices in GNNs. The author further utilizes LSH for improving the quality of sketches.

**Questions:**

See above

**Limitations:**

See above

**Strengths And Weaknesses:**

Strengths:

1. To the best of my knowledge, the sketing method proprosed in this paper is novel. I really like the idea of propagating the sketches instead of merely storing sketched activations in the forward pass.

2. The proposed techinique can be applied to different kinds of GNNs.

3. The theoretical analysis is correct and insightful, although I would like to note that it has a gap to the empirical results.

Weaknesses

1. I suggest the author tone down the claim "Sketch-GNN is the first sketch-based training algorithm for GNNs" and discuss the difference to the similar work EXACT [1].
In fact, [1] already applied the random projection-based sketching to the node embeddings.
If you take a close look to Fig. 1 in [1], the main difference of sketching between this work and [1] are:
(1) SketchGNN applied the sketch during the forward pass and propagate the sketched node embeddings layer-by-layer, while the sketching in [1] only affect the backward pass since it use the original note embeddings for propagation during the forward pass.
(2) SketchGNN and EXACT sketch the note embedding along different axis.
[1] estimates $CXW$ as $CXW\approx C(XW)R_1R_1^T$, where $R_1\in\mathbb{R}^{d\times r}$ is the sketching matrix with $c\leq d$.
For Sketch-GNN, it estimates $CXW$ as $CXW\approx CR_2R_2^T(XW)$, where $R_2\in\mathbb{R}^{c\times n}$.
However, [1] cannot accerlate the training process because they cannot propagate the sketched embeddings like SketchGNN.
If I was wrong, I would be more than happy be corrected.

2. Based on the weak1, I checked the performance gap between [1] and this work, I found that [1] has much less accuracy drop (e.g., SKetchGNN has >1% accuracy drop on the ogbn-arxiv, while [1] is less than 0.5%). My guess is that because this work introduces errors and gets accumulated as it passes layer-by-layer during the forward pass. Theorem 1 in this work is actually indirect to the optimization process in the sense that it does not charactize the gradient dynamics. In fact if you apply any sketching technique during the forward pass, then the gradient is provably biased [2] (see Eq. 11 in [2] and its surrounding explanation).
So I suggest discussing the gradient bias into the theoretical analysis, which is not very hard to do, but will improve the soundness of this work.

3. The overall algorithm is too complicated in the sense that it has at least four additional hyperparameters. (1) The truncation order of the power series for appoximating the activation function. (2) sketch ratio $c/n$. (3) the subset size $B$ for LSH. (4) The update intervel $T_{LSH} $for LSH. Too many additional hyperparameters highly limit the practical usage of the method because the cost of extra hyperparameter tuning grows exponentially.

4. (important) What do you mean by "train memory" in Table 1? From L368, it seems to be the peak memory usage during training. I checked the codebase which is based on PyG. I believe the reported data is wrong in the sense that the full-batch GNN baseline on ogbn-arxiv should roughly requires 1G+ memory (roughly 700MB for activations) to train [3].

5. What do you mean by "epoch time"? Is it the time for training one epoch? For GraphSAINT and VQ-GNN, each epoch contains serveral iterations while the full graph counterpart only have one iteration per epoch. Could you make this part more clear? Also, what is the epoch time of SketchGNN versus GraphSAINT/VQ-GNN on ogbn-products?

6. The authors do not discuss about the compatibility of BatchNorm and Dropout. I believe them can be applied to the sketched note embeddings.
However, I do not think they are useful in this case, especially for Dropout because sketched embeddings are denser compared to the original one.

7. One last question is about the activations.
I pretty like the idea of overcoming the non-linearity by approximating the activation function as a power series. Besides Sigmoid, could the author do an ablation study for approximating ReLU function using learned coefficents (since we almost do not use Sigmoid as the activation function in GNNs)?

typos: L234 ldots.

At last, I would like to say I like the idea of this paper and I think the idea is elegant. If my questions are discussed properly, I will raise my score.
-----Update----
After reading the rebuttal, most of my questions are addressed accordingly. I would like to raise my score to 7.

[1] EXACT: SCALABLE GRAPH NEURAL NETWORKS TRAINING VIA EXTREME ACTIVATION COMPRESSION

[2] Faster Neural Network Training with Approximate
Tensor Operations

[3] GNNAutoScale: Scalable and Expressive Graph Neural Networks via Historical Embeddings

---

> ### Author Response · Authors · 2022-08-02
> **$\newcommand{wwt}{\textcolor{blue}{\mathrm{wwt9}}}$Response to Reviewer $\wwt$ (1/3)**
>
> $\newcommand{wwt}{\textcolor{blue}{\mathrm{wwt9}}}$
> We appreciate Reviewer $\wwt$'s feedback and review. We are encouraged that Reviewer $\wwt$ likes our idea and finds our contribution novel and theoretical analysis insightful. Reviewer $\wwt$ raises some questions regarding comparison to a recent related work and understanding of some of the experiment results (Q2). We would like to address Reviewer $\wwt$'s questions below.
>
> ---
>
> > Q1: **Related Work: EXACT**: *I suggest the author tone down the claim "Sketch-GNN is the first sketch-based training algorithm for GNNs" and discuss the difference to the similar work EXACT [1]. In fact, [1] already applied the random projection-based sketching to the node embeddings. If you take a close look at Fig. 1 in [1], the main difference in sketching between this work and [1] are: (1) SketchGNN applied the sketch during the forward pass and propagated the sketched node embeddings layer-by-layer, while the sketching in [1] only affect the backward pass since it uses the original note embeddings for propagation during the forward pass. (2) SketchGNN and EXACT sketch the note embedding along a different axis. [1] estimates $CXW$ as $CXW\approx C(XW)R_1R_1^T$, where $R_1\in\mathbb{R}^{d\times r}$ is the sketching matrix with $c\leq d$. For Sketch-GNN, it estimates $CXW$ as $CXW\approx CR_2R_2^T(XW)$, where $R_2\in\mathbb{R}^{c\times n}$. However, [1] cannot accelerate the training process because they cannot propagate the sketched embeddings like SketchGNN. If I was wrong, I would be more than happy to be corrected. [1] EXACT: Scalable Graph Neural Networks Training via Extreme Activation Compression*
>
> We really appreciate Reviewer $\wwt$ for pointing us to this recent work, EXACT, and we think it is very relevant to our paper. First of all, we agree to revise our claim in Line 67 to "Sketch-GNN is the first sub-linear complexity training algorithm for GNNs based on LSH and tensor sketching." After reading the EXACT paper, we completely agree with the three differences between Sketch-GNN and EXACT as summarized in the question:
> 1. **Sketch-GNN enjoys sub-linear training complexities while EXACT does not**. We want to address that Sketch-GNN and EXACT are aiming for very different goals; Sketch-GNN is sketching the graph to achieve sub-linear complexity, while EXACT is sketching to save the memory footprint of non-linear activations.
> 2. Sketch-GNN sketches the graph size dimension while EXACT sketches the feature dimension.
> 3. Sketch-GNN propagates sketched representations while sketching in EXACT only affects the back-propagation. **EXACT cannot accelerate the training process and may even introduce time overhead**.
>
> > Q2: **Gradient Estimation Error**: *Based on the weak 1, I checked the performance gap between [1] and this work, and I found that [1] has a much less accuracy drop (e.g., SKetchGNN has >1% accuracy drop on the ogbn-arxiv, while [1] is less than 0.5%). My guess is that it is because this work introduces errors and gets accumulated as it passes layer-by-layer during the forward pass. Theorem 1 in this work is actually indirect to the optimization process in the sense that it does not characterize the gradient dynamics. In fact, if you apply any sketching technique during the forward pass, then the gradient is provably biased [2] (see Eq. 11 in [2] and its surrounding explanation). So I suggest discussing the gradient bias in the theoretical analysis, which is not very hard to do but will improve the soundness of this work. [2] Faster Neural Network Training with Approximate Tensor Operations*
>
> As explained in the answer for Q1 above, Sketch-GNN and EXACT are different algorithms achieving different goals, although they both applied sketching techniques in GNNs. In this regard, we believe only comparing their reported performance drop is not necessarily fair/appropriate.
> * Firstly, their training efficiency is different. EXACT only reduces the memory footprint of non-linear activations but introduces some time overhead, while Sketch-GNN reduces both training memory and time to be sub-linear w.r.t. the graph size.
> * Secondly, since they are sketching along two different axes, the errors are actually from two different sources, i.e., sketching graph in Sketch-GNN and sketching features in EXACT.
>
> We agree that the bias of gradient estimation should be discussed. However, the Equation 11 of [2] is discussed under the column-row sampling (CRS) algorithm instead of the count/tensor sketch algorithm used by Sketch-GNN. Further work is needed to characterize the error of gradient estimation theoretically. We will try to add related discussions to the appendices in future updates.

---

> > ### Author Response · Authors · 2022-08-02
> > **$\newcommand{wwt}{\textcolor{blue}{\mathrm{wwt9}}}$Response to Reviewer $\wwt$ (2/3)**
> >
> > > Q3: **Many Hyper-Parameters**: *The overall algorithm is too complicated in the sense that it has at least four additional hyperparameters. (1) The truncation order of the power series for approximating the activation function. (2) sketch ratio $c/n$. (3) the subset size $B$ for LSH. (4) The update interval $T_{LSH}$ for LSH. Too many additional hyperparameters highly limit the practical usage of the method because the cost of extra hyperparameter tuning grows exponentially.*
> >
> > Thanks for this question. We believe none of these four hyper-parameters need time-costly tuning to optimize the performance. Given an input graph of size $n$, the sketch ratio $c/n$ is roughly determined by the resource limit, e.g., how much GPU memory is available since the performance is usually monotonically increasing w.r.t the sketch ratio. Setting the truncation order $r=3$ is justified by theory (see Line 200) and usually works well in practice. We also found the performance of Sketch-GNN is insensitive to the subset size $B$ (given it is not too small) and the LSH update interval $T_{LSH}$ (because of the slow change phenomenon of LSH, see Line 353 and Figure 1b). Therefore, in practice, we do not require an exhaustive grid search to tune these four hyper-parameters.
> >
> > > Q4: **Train Memory**: *What do you mean by "train memory" in Table 1? From L368, it seems to be the peak memory usage during training. I checked the codebase, which is based on PyG. I believe the reported data is wrong in the sense that the full-batch GNN baseline on ogbn-arxiv should roughly require 1G+ memory (roughly 700MB for activations) to train [3].*
> >
> > We want to apologize that the peak training memory of full graph training on ogbn-arxiv is wrong, and we have corrected it. The correct peak memory is 983MB, and this number can be easily reproduced by running the `gnn.py` script in [the OGB GitHub repository](https://github.com/snap-stanford/ogb/tree/master/examples/nodeproppred/arxiv) with hyper-parameter `--hidden_channels=128` and `--dropout=0` and set `cached=False` for `GCNConv` layers. We use the `torch.cuda.max_memory_allocated()` function to report the peak memory usage. The problem with our previous code snippet for full graph training is that we placed `torch.cuda.max_memory_allocated()` at an earlier location, and only a part of the training memory is counted. We use independent code snippets for different baselines and implementation of our algorithm. We have verified this problem does not occur for the other reported memory usages for the other baselines.
> >
> > > Q5: **Epoch Time**: *What do you mean by "epoch time"? Is it the time for training one epoch? For GraphSAINT and VQ-GNN, each epoch contains several iterations, while the full graph counterpart only has one iteration per epoch. Could you make this part more clear? Also, what is the epoch time of SketchGNN versus GraphSAINT/VQ-GNN on ogbn-products?*
> >
> > Thanks for this question. The epoch time refers to the time to do one complete pass of the training dataset (or its sketched/compressed counterpart) through the training algorithm. GraphSAINT and VQ-GNN are mini-batch training algorithms, and there are multiple iterations per epoch. In contrast, "full-graph training" (oracle), graph coarsening, and sketch-GNN have only one iteration per epoch. Although the number of iterations per epoch is not the same, the time to do one complete pass of the dataset (i.e., the epoch time) is a fair measure of the training time complexity.

---

> > > ### Author Response · Authors · 2022-08-02
> > > **Response to Reviewer $\wwt$ (3/3)**
> > >
> > > $\newcommand{wwt}{\textcolor{blue}{\mathrm{wwt9}}}$
> > > > Q6: **Compatibility with BatchNorm and Dropout**: *The authors do not discuss the compatibility of BatchNorm and Dropout. I believe they can be applied to the sketched note embeddings. However, I do not think they are useful in this case, especially for Dropout, because sketched embeddings are denser compared to the original one.*
> > >
> > > Thanks for this interesting question. If we want to make sketch-GNN compatible with BatchNorm and Dropout, we have to carefully design approximated update rules for these layers, just like what we have done for GNN layers. Whether such approximated update rules on the sketched representation exist is a non-trivial question and is somehow beyond the scope of this paper (since this paper focus on sketching GNN layers). On the other hand, we also doubt whether BatchNorm and Dropout are helpful in the sketched representation. It is likely they introduce more errors compared with the regularization effects.
> > >
> > > > Q7: **Activation**: *One last question is about the activations. I pretty much like the idea of overcoming the non-linearity by approximating the activation function as a power series. Besides Sigmoid, could the author do an ablation study for approximating the ReLU function using learned coefficients (since we almost do not use Sigmoid as the activation function in GNNs)?*
> > >
> > > Yes, we can approximate the ReLU activated representation using the polynomial tensor sketch (PTS) with learned coefficients under the same setup as Figure 1a. At sketch ratio $c/n=0.1$, the relative error of PTS with learned coefficients is $3.4\times10^{-2}$, still reasonably small and comparable to the related error of core-set method, $2.2\times10^{-2}$. We choose to use Sigmoid activation as the ground truth in Figure 1a because Sigmoid is differentiable and has Talyor expansion at 0, so we can evaluate the gain of learning the polynomial coefficients. However, it does not mean we can only approximate the Sigmoid Activation well.
> > >
> > > ---
> > >
> > > We again thank Reviewer $\wwt$ for the time and effort in reviewing our paper. We hope that our explanations above have addressed all concerns of Reviewer $\wwt$. If so, we would greatly appreciate it if Reviewer $\wwt$ could consider raising their score. We are happy to answer any further questions.

---

> ### Comment · Reviewer_wwt9 · 2022-08-07
> **Responses to the authors**
>
> Thank you for your detailed replies. I have read the responses and most of my questions are addressed. I will raise my score accordingly. Again, I really like the idea of propagating sketched activations and I think the authors did the great work on exploring this direction:)

---

### Official Review · Reviewer_9WyJ · 2022-07-08

**Rating:** 7
**Confidence:** 3
**Soundness:** 3 good
**Presentation:** 3 good
**Contribution:** 3 good

**Summary:**

Scaling GNNs to large graphs is an important area of research. This paper proposes Sketch-GNN, which uses polynomial tensor sketch (PTS) and local sensitive hashing (LSH) to sketch the graph convolution operations in GNNs with sublinear training complexity. To be specific, PTS is used to estimate the non-linear GNN update rule, and LSH is used as a data-dependent method for learnable sketches.

Experiments on several graph datasets (ogbn-arxiv, Reddit and ogbn-product) proves the low memory and training time complexity of the proposed sketch-GNN method, which could achieve comparable performance with respect to other strong benchmarks including GraphSAINT and VQ-GNN. Ablation tests over sketching ratio c are also provided.

**Questions:**

Based on my understanding, sketch-GNN achieve sublinear training complexity because the sketch ratio c/n decreases as the graph size n increases. Therefore, as mentioned in the second bullet point in weakness, it would be great to see more empirical results that could support sublinear complexity.

**Limitations:**

Yes.

**Strengths And Weaknesses:**

Strength:

* This paper is well-written, and provides a nice way to incorporate tensor sketch and LSH into GNNs, which might inspire future work in this area.

* Some adaptations are incorporated to boost the performance of the proposed algorithm, including learnable coefficients for the non-linear activation function and data-dependent hashing with learnable LSH.


Weakness:

* As shown in table2 and table3, GraphSAINT uses comparable train memory (but longer epoch time), and could achieve better performance than sketch-GNN on average. So is the proposed sketch-GNN method good enough as an alternative for GraphSAINT?

* As mentioned in line 398-399, the needed sketch ratio c/n for sketch-GNN to achieve competitive performance reduces as graph size grows. However, only 1 sketch ratio are reported for each graph dataset. It would be nice to see results from multiple sketch ratios for each dataset to confirm this statement.

---

> ### Author Response · Authors · 2022-08-02
> **$\newcommand{WyJ}{\textcolor{green}{\mathrm{9WyJ}}}$Response to Reviewer $\WyJ$ (1/1)**
>
> $\newcommand{WyJ}{\textcolor{green}{\mathrm{9WyJ}}}$
> $\newcommand{HVXV}{\textcolor{red}{\mathrm{HVXV}}}$
> We appreciate Reviewer $\WyJ$'s feedback and review. We are encouraged that Reviewer $\WyJ$ likes the way we incorporate tensor sketch and LSH into GNNs. Reviewer $\WyJ$ raises some questions about the understanding of some empirical results, some of which we think may be due to misinterpretation. We would like to address Reviewer $\WyJ$'s questions below.
>
> > Q1: **How Sketch-GNN compares with GraphSAINT**: *As shown in table2 and table3, GraphSAINT uses comparable train memory (but longer epoch time), and could achieve better performance than sketch-GNN on average. So is the proposed sketch-GNN method good enough as an alternative for GraphSAINT?*
>
> Thanks for this insightful question.
> * GraphSAINT is a graph sampling method that enables training on a mini-batch of subgraphs instead of on the large input graph. GraphSAINT is easily applicable to any graph neural network (GNN), introduces minor overheads, and usually works well in practice.
> * However, **GraphSAINT is not a sub-linear training algorithm**. We have to iterate through the full batch of subgraphs in an epoch, and the training time complexity is still linear in the graph size.
> * In contrast, our proposed **Sketch-GNN is an approximated training algorithm of some GNNs with sub-linear time and memory complexities**. Sketch-GNN has the potential to scale better than GraphSAINT on larger graphs.
> * Besides, as a sketching algorithm, Sketch-GNN is suitable for some scenarios, for example, sketching big graphs in an online/streaming fashion.
> * Sketch-GNN can also be combined with subgraph sampling to scale up to extremely large graphs. Sketching the sampled subgraphs (instead of the original graph) avoids the decreasing sketch ratio when the input graph size grows to extremely large while with a fixed memory constraint. We validated this idea, and the details can be found in the answer to Q8 of Reviewer $\HVXV$.
>
> These important differences between Sketch-GNN and graph sampling methods like GraphSAINT are highlighted in section 4 and added to the appendices.
>
> > Q2: **Multiple Sketch-Ratio per Dataset**: *As mentioned in Line 398-399, the needed sketch ratio c/n for sketch-GNN to achieve competitive performance reduces as graph size grows. However, only one sketch ratio is reported for each graph dataset. It would be nice to see results from multiple sketch ratios for each dataset to confirm this statement. Based on my understanding, sketch-GNN achieve sublinear training complexity because the sketch ratio c/n decreases as the graph size n increases. Therefore, as mentioned in the second bullet point in weakness, it would be great to see more empirical results that could support sublinear complexity.*
>
> We want to clarify that we indeed use multiple sketch ratios in Table 3. And from Table 3, we can see the needed sketch ratio $c/n$ for Sketch-GNN to achieve competitive performance reduces as graph size grows since using a sketch ratio of 0.4 on ogbn-arxiv (169K nodes, 1.2M edges), 0.3 on Reddit (233K nodes, 11.6M edges), and 0.2 on ogbn-product (2.4M nodes, 61.9M edges) can achieve comparable performance to the baselines. In general, we think the results in Table 3 already answered this question.
>
> ---
>
> We again thank Reviewer $\WyJ$ for the time and effort in reviewing our paper. We hope that our explanations above have addressed all concerns of Reviewer $\WyJ$. We are happy to answer any further questions.

---

> > ### Comment · Reviewer_9WyJ · 2022-08-10
> > **Response to the authors**
> >
> > Thanks for your reply! Now I understand better the scalability difference between GraphSAINT and the proposed Sketch-GNN. I will raise my score accordingly.

---

### Official Review · Reviewer_Cdzo · 2022-07-10

**Rating:** 7
**Confidence:** 4
**Soundness:** 3 good
**Presentation:** 2 fair
**Contribution:** 3 good

**Summary:**

The authors propose a method to train graph neural networks more efficiently and hence make them amenable to large-sized graphs.
The main idea is to use hash maps on the graph nodes. More specifically, the authors propose to map the convolution matrices (n X n )and the feature matrix ( n X d ) to c hash buckets. Following this, the proposed algorithm only needs to access the c-dimensional sketched convolutional matrices and feature vectors, this leading to sub-linear complexity. The authors first present a framework based on count sketch and tensor sketches and derive feature update equations that only depend on the sketched data. However, using these updates in a straight-forward manner leads to instability during training. To address this, the authors propose to use locality sensitive hashing (LSH) and develop a practical algorithm for sketching based GNN training. The proposed algorithm then studies for its efficiency and performance across a few different datasets and baseline comparisons are provided.
On average, the proposed algorithm seems to perform better compared to the one other sub-linear time baseline algorithm in terms of efficiency and performance. In terms of efficiency, the algorithm outperforms other non sub-linear algorithms and also has similar performance.

**Questions:**

I have already listed some suggestions in the "weaknesses" section above. Apart from those I have the following questions/ suggestion:

1. Section D.2 also warrants a more mathematical discussion. It would be great to see the equations for how the sketched attention coefficients are computed.



**Limitations:**

The authors have discussed the limitations of their work.

**Strengths And Weaknesses:**

Strengths:

1. The presented algorithm is interesting and the usage of LSH in the context of GNNs is novel. Although the presented ideas are based mainly on existing work on using LSH for deep learning ([8] and [9]), the present work differentiates itself by applying those ideas to develop a framework to train graph neural networks.

2. Sub-linear complexity is particularly attractive and the presented results show that the proposed algorithm is good at taking advantage of the redundancy across graph nodes by using hash maps. The advantages of the algorithm in terms of efficiency and training are clearly shown in the experiments provided.

3. The paper is technically sound and has a good flow of ideas.

Weaknesses:

1. One of my main concerns is the presentation of the paper. In my opinion, the main contribution of the paper is a discussion of how LSH can be used to train GNNs (as opposed to using tensor sketching, which has been allocated a lot of real estate in the paper). On many occasions, the details of the main algorithm are deferred to the appendix. For example, the paper would greatly benefit from having Algorithm 1 in the main body. In my opinion, this algorithm and the other details presented in section C are more important than the presented error bounds in section 3.2 (which could have been in the appendix instead).

2. My second concern is regarding some of the experimental results:
      a. In Fig 1a, why were the weights chosen to be identity matrices? More elaborate experiments with the learnt weights will make these
          results more compelling.
      b. Table 1. should include results for GAT as well. In general, GATs are more likely to face complexity issues while training because of
           the expensive attention operation. I am curious to see the effect of the proposed algorithm on GATs.

3. Another observation is that on many occasions, the authors use verbal explanation of techniques rather than equations and a mathematical presentation. For example, lines 252-265 might be better explained using a series of equations. The same can be said about most other "subsections" presented in Section 3.3.

In general, the paper need some restructuring to address the following points:
a. provide more details of the actual algorithm in the main paper (especially for 'online learning of sketches' and 'avoiding O(n) in loss evaluation'
b. The presented error bounds are not very informative and could be moved to the appendix.

4. The paper could include more references for related work, given that the field of sampling and sketching for graph neural networks is quite well-studied. See for example:
[1] Calandriello, Daniele, et al. "Improved large-scale graph learning through ridge spectral sparsification." International Conference on Machine Learning. PMLR, 2018.
[2] Zheng, Cheng, et al. "Robust graph representation learning via neural sparsification." International Conference on Machine Learning. PMLR, 2020.
[3] Srinivasa, Rakshith S., et al. "Fast graph attention networks using effective resistance based graph sparsification." arXiv preprint arXiv:2006.08796 (2020).

---

> ### Author Response · Authors · 2022-08-02
> **$\newcommand{Cdzo}{\textcolor{orange}{\mathrm{Cdzo}}}$ Response to Reviewer  $\Cdzo$ (1/2)**
>
> $\newcommand{Cdzo}{\textcolor{orange}{\mathrm{Cdzo}}}$
>
>
> We appreciate Reviewer $\Cdzo$'s feedback and review. We are encouraged that Reviewer $\Cdzo$ finds the sub-linear complexity of the proposed algorithm attractive, the experiments convincing, and the techniques sound. Reviewer $\Cdzo$ raises some concerns on paper presentation and experimental details. We would like to address Reviewer $\Cdzo$'s questions below.
>
> ---
>
> > Q1: **Paper Presentation**: *In my opinion, the main contribution of the paper is a discussion of how LSH can be used to train GNNs (as opposed to using tensor sketching, which has been allocated a lot of real estate in the paper). On many occasions, the details of the main algorithm are deferred to the appendix. For example, the paper would greatly benefit from having Algorithm 1 in the main body. In my opinion, this algorithm and the other details presented in section C are more important than the presented error bounds in section 3.2 (which could have been in the appendix instead). In general, the paper needs some restructuring to address the following points: a. provide more details of the actual algorithm in the main paper (especially for 'online learning of sketches' and 'avoiding O(n) in loss evaluation' b. The presented error bounds are not very informative and could be moved to the appendix.*
>
> Thanks for the detailed suggestion. We agree that Algorithm 1 (currently in Appendix E) should be put in the main body if possible. However, we still think Theorem 1 is essential and should not be deferred to the appendices. Apart from the use of locality-sensitive hashing (LSH) in GNNs, another important contribution of our paper (as summarized in the conclusion section) is to approximate the non-linear activations in GNNs using the polynomial tensor sketch (PTS). Theorem 1 established an error-bound on the estimation error accumulated through multiple layers of GNNs and verifies the validity of sketching non-linear operations. Therefore we think Theorem 1 should be kept in the main body if possible. Due to the strict 9-page limit, it is hard to move Algorithm 1 to the main body right now, but we will adopt this suggestion for future versions. Apart from moving Algorithm 1, we refined Algorithm 1 and the relevant parts in section 3.3 (related to Q3 addressed below).
>
> > Q2: **Some Experimental Details**: *(a). In Fig 1a, why were the weights chosen to be identity matrices? More elaborate experiments with the learned weights will make these results more compelling. (b). Table 1. should include results for GAT as well. In general, GATs are more likely to face complexity issues while training because of the expensive attention operation. I am curious to see the effect of the proposed algorithm on GATs.*
>
> We appreciate these insightful questions. We answer them respectively as follows.
> 1. Question 2(a): Figure 1a is a comparison of the relative errors of sketching the non-linear unit $\sigma(CXW)$ using three different algorithms (core-set, Taylor expansion, learned from gradients (ours)). In Figure 1a, for the 'learned from gradients' method, we learn the polynomial coefficients $c_k$ of the polynomial tensor sketch (PTS) to optimize the approximation error w.r.t. the ground truth (i.e., $\sigma(CXW)$ when $\sigma(\cdot)$ is sigmoid). This is stated in Line 352. The experiments behind Figure 1a are designed to compare the approximation power of the sketching methods. If we optimize the PTS coefficients $c_k$ w.r.t. the classification loss (as in all the other experiments following Line 161), the PTS will no longer approximate a known activation, and we cannot evaluate the approximation error because we do not know the ground-truth $\sigma(CXW)$. In general, the experiments of Figure 1a have nothing to do with the downstream task, and that is why we do not optimize $W$ and fix it to the identity matrix. We are sorry for the confusion in understanding Figure 1a, and we have expanded the description in Line 352 to clarify the experimental settings in Figure 1a.
> 2. Question 2(b): For Sketch-GAT trained on ogbn-arxiv with one head, the pre-processing time is 38s, the epoch time is 0.42s, and the peak training memory is 92.7MB. For the GAT with the same architecture trained with GraphSAINT, the epoch time is 0.34s, and the peak training memory is 59.1MB. Although Sketch-GAT suffers from more expensive overhead to approximate the sketched attention units in each layer under this specific setting, Sketch-GNN enjoys sub-linear complexities while GraphSAINT does not.

---

> > ### Author Response · Authors · 2022-08-02
> > **$\newcommand{Cdzo}{\textcolor{orange}{\mathrm{Cdzo}}}$Response to Reviewer  $\Cdzo$ (2/2)**
> >
> > $\newcommand{Cdzo}{\textcolor{orange}{\mathrm{Cdzo}}}$
> > > Q3: **Explanation Style**: *another observation is that on many occasions, the authors use verbal explanation of techniques rather than equations and a mathematical presentation. For example, lines 252-265 might be better explained using a series of equations. The same can be said about most other "subsections" presented in Section 3.3.*
> >
> > We appreciate the suggestions. We agree that using more mathematical expressions can help make the presentation of section 3.3 more compact and concise. We refined Line 252-265 and some other parts in section 3.3.
> >
> > > Q4: **Additional References**: *The paper could include more references for related work, given that the field of sampling and sketching for graph neural networks is quite well-studied. See for example: [1] Calandriello, Daniele, et al. "Improved large-scale graph learning through ridge spectral sparsification." International Conference on Machine Learning. PMLR, 2018. [2] Zheng, Cheng, et al. "Robust graph representation learning via neural sparsification." International Conference on Machine Learning. PMLR, 2020. [3] Srinivasa, Rakshith S., et al. "Fast graph attention networks using effective resistance based graph sparsification." arXiv preprint arXiv:2006.08796 (2020).*
> >
> > Thanks for pointing these additional references to us. All three papers aimed to scale up graph learning ([1] for graph-Laplacian-based learning, [2] for general graph neural networks, and [3] for graph attention networks) with graph sparsification, i.e., removing task-irrelevant and redundant edges from the large input graph. Graph sparsification will not reduce the number of nodes; thus, the memory reduction of node feature representation is limited. However, some carefully designed graph sparsification may enjoy small approximation error (thus smaller performance drops) and improve the robustness of learned models. We added corresponding discussions to the extended related work section in Appendix G.
> >
> > > Q5: **Refine Appendix D.2**: *Section D.2 also warrants a more mathematical discussion. It would be great to see the equations for how the sketched attention coefficients are computed.*
> >
> > Thanks for the suggestion. We refined Appendix D.2 to make the description clearer.
> >
> > ---
> >
> > We again thank Reviewer $\Cdzo$ for the time and effort in reviewing our paper. We hope that our explanations above have addressed all concerns of Reviewer $\Cdzo$. We are happy to answer any further questions.

---

> > > ### Comment · Reviewer_Cdzo · 2022-08-07
> > > **Response to rebuttal**
> > >
> > > I thank the authors for providing a detailed response to my questions. I was able to better understand the reason behind using identity matrix as the weights for the experiments in Figure 1a. Most of my other concerns were regarding the presentation of the paper and not regarding the technical details, which are well formulated. I believe that the authors have sufficiently addressed these concerns. So I will increase my score.

---

### Meta-Review · Area_Chair_6UGe · 2022-08-31

**Recommendation:** Accept
**Confidence:** Certain

**Metareview:**

The authors propose the use of sketch GNN (based on compressing relevant matrices and sketching typical GNN operations via hashing) to enable better scaling of graph neural networks to very large graphs. The reviewers are all in favor of accepting the paper (with three accepts and one weak accept), and therefore I recommend its acceptance. I encourage the authors to take into account the reviewer comments, as they already indicated they will do during the rebuttal period, when preparing the camera ready version.

**Award:**

No

---

### Decision · Program_Chairs · 2022-09-14

Accept